# Climate risks, digital media, and big data: following communication trails to investigate urban communities' resilience

Rosa Vicari[1], Ioulia Tchiguirinskaia[1], Bruno Tisserand[2], and Daniel Schertzer[1]

[1]Hydrology Meteorology and Complexity Laboratory, École des Ponts ParisTech, Marne-la-Vallée, Champs–sur–Marne, 77455, France
[2]Veolia Research and Innovation, Chemin de la Digue, Maisons Laffitte, 78600, France

**Correspondence:** Rosa Vicari (rosa.vicari@enpc.fr)

**Abstract.** Nowadays, when extreme weather affects an urban area, huge amounts of digital data are spontaneously produced by the population on the Internet. These 'digital trails' can provide an insight on the interactions existing between climate-related risks and the social perception of these risks. According to this research 'big data' exploration techniques can be exploited to monitor these interactions and their effect on urban resilience. The experiments presented in this paper show that digital research can bring out the key issues in the digital media, identify the stakeholders that can influence the debate and, therefore, the community's attitudes towards an issue. Three corpora of Web communication data have been extracted: press articles covering the June 2016 Seine River flood; press articles covering the October 2015 Alpes-Maritimes flood; tweets on the 2016 Seine River flood. The analysis of these datasets involved an iteration between manual and automated extraction of hundreds of key terms; aggregated analysis of publication incidence and key term incidence; graph representations based on measures of semantic proximity (conditional distance) between key terms; automated visualisation of clusters through Louvain modularity; visual observation of the graph; quantitative analysis of its nodes and edges. Through this analysis we detected topics and actors that characterise each press dataset, as well as frequent co-occurrences and clusters of topics and actors. Profiling of social media users gave us insights on who could be the opinion makers" on Twitter. Through a comparison of the three datasets, it was also possible to observe how some patterns change over time, in different urban areas and in different digital media contexts.

*Copyright statement.* TEXT

## 1 Introduction

This paper presents a study on how digital media represent urban resilience during extreme weather and in the following weeks. The approach proposed here aims at exploiting the huge amount of Web data that are produced during and after a climate risk. Analysis of 'big data' corpora drawn from digital media has been already employed in research fields related to communication on climate risks. A series of studies analyse and compare the existing controversies on climate change in the web sphere (Niederer, 2013; Rogers and Marres, 2000; Chavalarias, 2015; climaps.eu, last access: 18 May 2018). Other

research works investigate the use of social media during crisis due to climate hazards (Palen et al., 2010; Morss et al., 2017; Bruns et al. 2012; Lanfranchi et al., 2014; Gaitan et al., 2014; J.C. Chacon-Hurtado, 2017).

The originality of our approach lies in the fact that it is framed in the context of research on urban resilience assessment. In this study we refer to resilience as the "the capacity of a system to absorb disturbance and reorganise while undergoing change so as to still retain essentially the same function, structure, identity, and feedbacks" (Walker et al., 2004, p. 2). This approach to resilience, defined by Walker et al. (2004) as "social-ecological resilience", has theoretical and practical implications (for a review, see Vicari et al., 2019).

Our research aims at contributing to the comprehension of the interactions that exist among different urban resilience drivers. According to Mangalagiu, one of the challenges of the 21st century is giving the attention to the 'complex and causal linkages between human, technological, environmental and global biophysical systems' (Mangalagiu et al., 2012, p. 2). In our view, quantifiable variables facilitate the investigation of the relations between different physic-environmental and socio-economic components of urban systems. Furthermore, quantitative indicators are helpful to cross-compare different locations and time points. A huge variety of quantitative indicators are proposed in the literature on different resilience assessment approaches (Cutter et al. 2008; 2010; UN/ISDR, 2008; Resilience Alliance, 2010; Keating et al., 2014). In this paper, the focus is put on those quantitative data that can be used to investigate the social representation of climate risks. Digital media are a source of quantitative information that can be automatically extracted and analysed through computer-aided exploration techniques such as advanced text mining and graph representation. A quantitative analysis of digital communication patterns easily leads to an evaluation of how different space and time variables affect these patterns. It also facilitates the analysis of how communication trends and other resilience drivers (e.g. an environmental factor) mutually influence each other. When these correlations exist, they are a necessary basis to understand how social perception of climate risks affects urban resilience.

Besides examining these methodological challenges, we also intend to contribute to the comprehension of the social perception of urban resilience to climate risks through digital media. We present an analysis of three datasets in Sect. 3, 4 and 5: the press articles covering the June 2016 Seine River flood; the press articles covering the October 2015 Alpes-Maritimes flood; the tweets on the 2016 Seine River flood. We discuss an initial analysis of the topics and actors mentioned in the media texts, as well as of the thematic subsets and term co-occurrences that characterise each dataset. We also compare the three datasets and reflect on how the debate changes over time, in different urban areas and in different media contexts.

## 2  Data

Online press articles and social media posts are second-hand data. Indeed, as it is discussed by Venturini (Venturini et al. 2014), the researcher cannot directly control the production of these data and he should question himself about their production context and process. For instance, news items publication follows a set of journalistic values, the so-called 'newsworthiness' (Boyd, 1994), that determine if and how much a story is important for a media outlet and its audience. An example of news value is 'the greater the drama, the greater its prominence in conversation': this kind of news, that is expected to get their audience talking, is considered more worthy than others. These values are translated in a hierarchy of news that will guide news

programming. In the case of social media, the problem of the digital divide (i.e. Internet access, skills and usage inequality) leads the researcher to consider the socio-demographic characteristics of social media users. For example, when analysing tweets, it should be taken into account that in France the population over 45 years is not well represented by a sample of Twitter users, while the population with a university degree is overrepresented. Indeed, in France in 2017, 16% of Twitter users were over 45 years old and 40% of the users had a university degree (excluding BTS[1]) (blogdumoderateur.com, last access: 18 May 2018)[2]. According to Venturini 'digital traces are not natural items but artefacts created in a specific environment and with specific objectives'. However, this doesn't reduce their value because their publication process can be a source of information on the social representation of reality, for instance the media representation of climate-related threats. Even though digital communications make possible a more direct observation of social phenomena, these data need to be contextualised and interpreted.

## 2.1 Press articles on the 2016 Seine River flood

During the night of 3 June 2016, the Seine River reached 6,1 meters, the highest discharge on record since 1982. We picked the Seine River flood that occurred in June 2016 as a case study because of its prominent media impact. According to a search of French press articles on Europresse (europresse.com, last access: 18 May 2018), on the 3rd of June 2016 the press coverage of this flood event reached a peak of 310 articles published in one day (corresponding to 29 437 terms, as illustrated by Fig. 1a). This is a remarkable figure considering that the same French media published 591 articles in one day on Trump's victory on the 8th of November 2016. Media visibility influences public opinion, hence stakeholders' attitudes towards risks and disasters, and related resilience policies or projects[3]. Therefore this flood event is worth exploring from the urban resilience perspective.

The corpus of articles on the Seine River flood consists in 761 documents selected on the basis of the following criteria: French press articles published from 15/05/2016 to 15/10/2016, with a title including the terms ('crue' or 'inond*') and ('Seine' or 'Île-de-France' or 'Paris' or 'Région Parisienne'). The selection starts from the day of publication of the very first press articles on the 2016 Seine River flood. We chose the duration of five months in order to obtain a corpus of press articles that were large enough for graph representation and because the first analysis was carried out in November 2016, five months after the Seine River flood of June 2016.

---

[1]BTS (Brevet de Technicien Supérieur) is a French diploma of higher education that is obtained after two or three years of studies in a highly specialised field.

[2]According to the French National Institute of Statistics and Economic Studies (insee.fr, last access: 20/01/2019), in 2017, in France, 20,9% of the population between 25 and 65 years old had a university degree (obtained after more than two years of university studies) and 41% of the population was over 45 years old.

[3]Media contribute to our perception of reality (including risks and disasters) through selection and omission of information. For example, the UK government reassurance campaign contributed to spreading the mad cow disease, which resulted in millions of slaughtered animals and the deaths of 226 people. Furthermore, humans respond more forcefully to emotional appeals than to facts like in the case of the Indian Tsunami earthquake (2004): images and stories from tourists and the extreme language used by the media led to a higher donors' response compared to other disasters with more victims.

## 2.2 Press articles on the 2015 flood in the Alpes-Maritimes Department

On the 3$^{rd}$ and 4$^{th}$ of October 2015 extreme rainfall caused river floods in the Alpes-Maritimes Department. Cannes, Antibes, Vallauris, Biot and Mandelieu-la-Napoule were the most affected municipalities. The press coverage of this flood event was more limited (286 articles over five months) in comparison to the Seine River flood (761 articles over five months), even though the first flood took a huge toll on human life with 20 deaths.

The corpus of articles on the Alpes-Maritimes flood includes 286 documents. We used the following criteria for the selection: French press articles from 15/09/2015 to 15/02/2015, with a title including the terms ('crue' or 'inond*') and a reference to at least one of the locations affected by the flood[4]. The selection starts from the day of publication of the very first press articles on the 2015 Alpes-Maritimes flood. We decided to consider the same duration (five months) in the analysis of the first and the second press article corpora in order to facilitate a cross comparison between the two case studies.

## 2.3 Tweets on the 2016 Seine River flood

As mentioned above, the press select and prioritise the news worthy information, hence it defines the prominent topics and their organisation in thematic clusters. In this way, editors and journalists obviously influence the public perception of an extreme weather event, even though a two-way relationship exists between the press and the audience. In the digital age, the role of editors and journalists as exclusive news mediators has been progressively fading. Indeed, access to information has hugely increased in terms of variety and quantity, as a consequence of different factors, among others the development of public relations by non-journalistic organisations and the pervasive role of the Web sphere (Bucchi, 2013; Trench, 2008). In this context, a corpus of texts published on a social media deserves to be analysed and compared to the press articles corpus in order to have insight of the public perception of a flood event beyond the borders of the journalistic arena. The third dataset analysed in this paper is a corpus of tweets covering to the Seine River flood of June 2016. The choice to focus on Twitter (twitter.com, last access: 18 May 2018) is due to the fact that public authorities and citizens are increasingly using Twitter during natural disasters as a two-way early warning and information channel. According to Bruns and Liang (2012, p. 1), Twitter is particularly suitable for crisis communication, indeed with its 'flat and flexible communicative structures' any visitor can access public tweets: users that are not yet followers of the account that disseminates the information on a crisis event or even visitors that are not registered on Twitter. Furthermore, hashtags can be used by any visitor to search for tweets on a specific topic. Such communicative structure facilitates fast, large-scale collection of information.

The corpus of tweets was selected on the basis of two criteria. The first one is the time span: all tweets were published from 28/05/2016 to 2/7/2016. We decided to consider one month duration because during the flood and in the following month

---

[4]The titles of the articles selected for the second case study have a title referring to at least one of the following locations: ("Alpes-Maritimes" or "Cannes" or "Antibes" or "Vallauris" or "Biot" or "Mandelieu-la-Napoule" or "Bouches-du-Rhône" or "Var" or "Vaucluse" or "Drôme" or "Siagne" or "Brague" or "Fréjus" or "Reyran" or "Vallauris-Golfe-Juan" or "Cagnes-sur-Mer" or "Le Cannet Mougins" or "Nice" or "Roquefort-les-Pins" or "La Roquette-sur-Siagne" or "Théoule-sur-Mer" or "Valbonne" or "Villeneuve-Loubet" or "Les Arcs" or "Brignoles" or "Cabasse" or "Callas" or "Camps-la-Source" or "Flassans-sur-Issole" or "Côte d'Azur" or "sud-est" or "Flayosc" or "Forcalqueiret" or "Fréjus" or "Méounes-lès-Montrieux" or "La Motte" or "Néoules" or "Puget-sur-Argens" or "La Roquebrussanne" or "Saint-Antonin-du-Var" or "Saint-Raphaël" or "Le Thoronet" or "Trans-en-Provence").

the incidence of tweets was relevant and the portion of tweets referring to other floods (outside the Seine River Basin) was reduced. The second criterion was the presence of relevant hashtags: each tweet contains at least one of these hashtags '#crue', '#crueparis', '#crueseine', '#inondation', '#inondations', '#pluies', '#Seine'. As it is suggested by Bruns and Liang (2012), thanks to hashtags, it is possible to focus on those tweets where terms related to a flood are marked as important information. Geolocation was not used as a criterion because the sample of tweets would have been very small: few users provide such detail with their tweet (Bruns et Liang, 2012). We first obtained 10 073 tweets, the corpus was then refined and reduced to 4497, after deleting duplicates and the tweets that included the term #crue but referred to 'Motley Crue', 'uncooked food' or 'cruelty' (in French). In order to facilitate a comparison with the two previous corpora, the tweets that mentioned locations outside the Paris region were deleted as well.

## 3 Methodology

Stakeholders' perception of a controversial issue is a community characteristic (and a social impact when change occurs) that can be analysed through surveys, meetings and interviews. Surveys provide information on population attitudes at aggregated level, while interviews and meetings (e.g. focus groups) provide insights about how and why particular attitudes are developed at individual level or at small-group level. However, big data exploration techniques make possible to get beyond the dichotomy between the aggregated structure and the individual component when studying social connections (Latour, 2012). The following examples of analysis of digital texts illustrate how computer aided exploration techniques can be employed to gain insight both on the intensity and the quality of Web communications. Indeed, thanks to an automated big data exploration tool such as Gargantext V2 (Chavalarias and Delanoe, 2017), it is possible to quickly navigate through huge masses of digital information, follow the connections among cultural contents (e.g. articles, blog posts, tweets), popular topics, the names of public figures or organisations.

A multitude of methods have been developed in the field of the scientometrics (Leydesdorff and Milojevic, 2012) to reveal the internal structure of cultural domains, through an automated analysis of data extracted from digital texts (such as authors, sources, documents, citations, links, references, terms, etc.). One of these methods, employed by Gargantext, is 'co-word analysis', a technique to map the semantic structure in a database that exploits the statistics on the frequency of co-occurrences between pairs of terms. We chose to use this open-source software to facilitate the replicability of the study and because Gargantext is unique in terms of ergonomics (see Supplement S1 for further details on how it works and on the required manual interventions). Furthermore, thanks to the collaboration between the HM&Co laboratory and ISC-PIF we benefited of support from the developers of Gargantext.

With Gargantext we can extract, automatically as well as manually, a list of key terms from a corpus of texts. This list of terms is then used by Gargantext to compute a graph representation. A 'graph' or a 'network' 'is a simplified abstract representation of a system, such as text database, that is meant to facilitate a smart navigation in the analysed system. It is a mathematical structure that represents a collection of interconnected objects. The objects are called 'nodes' (or 'vertices') and the connections are called 'edges' (or 'links') (Newman, 2010). The analysis of the graph leads to the comprehension of its single objects (i.e.

nodes), the interactions (i.e. edges) among those components and the pattern of interactions (i.e. graph). In the graphs computed with Gargantext the nodes represent the key terms and the edges represent a co-occurrence relation between two terms. In this research we computed graph representations on the basis of the semantic proximity measure, between pairs of key terms, that is called 'conditional distance'. Conditional distance between two terms is the probability that these terms co-occur in the same meaning unit (e.g. a press article). Gargantext graphs are weighted: a value (called the 'weight' of the edge) from zero to one is assigned to each edge of the graph and it indicates the probability that two terms co-occur. Furthermore, Gargantext can display the degree of each node, a measure that indicates the number of edges connected to each node. The nodes are assembled in cohesive subsets (or clusters) through a clustering algorithm, more specifically through Louvain modularity. Each cluster corresponds to a group of key topics and key actors that frequently appear in the same article. Gargantext highlights with the same colour all the terms belonging to the same cluster.

### 3.1 Methodology implemented to analyse the press articles

#### 3.1.1 Dataset extraction

The two corpora of articles were extracted through Europresse.com, a press archive that is accessible on the Internet by subscription. Articles can be selected on Europresse.com on the basis of keywords (in the title or in all the article), authors' name, language, the type of media (frequency of distribution, geographical area of distribution, language, country), media name, publication dates. Europresse has the advantage that it is possible to export the press articles in a format that is compatible with Gargantext. Furthermore, Europresse gives access to press sources in different languages (unlike Argus de la Presse, another important press online archive). This characteristic opens the path to future research applied to other countries than French-speaking countries.

#### 3.1.2 Aggregated analysis

A first histogram was created to illustrate the intensity of press coverage, expressed as the number of published terms per day, and how it evolved over one month. The following step of the analysis consisted in selecting terms that correspond to a range of 'flood resilience solutions'. 'Flood resilience solutions' is understood here in a large sense, as any kind of solution that is implemented to reduce flood damages. Since the objective of the analysis was to comprehend how Paris resilience is represented by the media, we selected all the terms referring to solutions to cope with flood risk.

For each corpus, a first list was automatically established by Gargantext algorithms. These lists were then manually refined on the basis of the thematic relevance of each term (as a solution aimed to cope with a flood event) and finally merged. This resulting list (Tab. S2.1, translated in Tab. S2.2) was used to analyse the occurrence of the key terms in each corpus, except for the terms with fe wer than five occurrences that were not considered in order to highlight the most frequent topics.

We used the key terms list to compare the total number of published terms per day with the number of published key terms per day referring to the topic of flood resilience solutions. The comparison between the overall corpus with the sub-group of key terms aimed to monitor how the quality of the content evolved over time. Further insights can be obtained through a

comparison between the two histograms based on different corpora of press articles and with the histogram based on a corpus of tweets.

### 3.1.3 Graph representation

After analysing the key terms incidence in an aggregated manner, the second step of the analysis was aimed at representing a complex communication field and revealing the pattern of interactions that exist between different topics and public figures or organisations. The graph was based on an adjusted version of the key term list referring to flood resilience solutions. Indeed, for a better comprehension of the context, where these solutions were implemented, we added to the initial list the terms referring to Paris infrastructure and properties. Since another objective of this analysis was to identify the opinion makers, we also selected terms referring to stakeholders. Furthermore, synonyms ('inhabitants' and 'residents'), declensions of terms (e.g. 'scientist' and 'scientists') and equivalent forms (e.g. 'Établissement Public Seine Grands Lacs', the public institution managing the Seine River Basin, and 'EPTB Seine Great Lacs') were merged. Once the key term list was refined, a graph was generated on the basis of the conditional distance between pairs of key terms.

### 3.1.4 Visual observation of the graph

As it is suggested by Venturini (Venturini et al., 2014), we based the first analysis of the results on a visual observation of the graph and in particular of its clusters: which nodes belong to the same cluster and what the corresponding topics and actors are. According to Venturini's approach, each cluster can be associated with a macro-theme, chosen by the researcher, corresponding to an expression that sums up the terms in the cluster. In our research the macro-themes correspond to resilience management areas.

### 3.1.5 Quantitative analysis of the nodes and the edges

Further information on the graph can be obtained through a quantitative analysis: the values corresponding to the node degrees and to the edge weights can be easily extracted through Gephi, a graph visualisation software (gephi.com, last access: 18 May 2018), and then compared. Gephi has the advantage of being an open-source software and of opening GEXF files, the format of the graph files produced by Gargantext. Furthermore, with Gephi it is possible to easily convert GEXF files in two Excel tables with the node degrees and the edge weights. With Excel it is then possible to order these values in descending order and generate the figures in Supplement S3.1, S3.2, S4.1, S4.2.

## 3.2 Methodology implemented to analyse the tweets on the Seine River flood

### 3.2.1 Extraction of the dataset

The corpus of tweets covering the Seine River flood of June 2016 was selected through 'Twitter Advanced Search' (twitter.com/search-advanced, last access: 8 July 2018), a free and ergonomic service provided by Twitter. We then used the scraping tool Dataminer (an open-source Chrome extension software) to convert HTML data, that appear in the browser window, into clean Excel table

format. By combining these two tools, it is possible to extract Twitter data, even after their publication and without any rate limit. An example of a different method to extract tweets is Twitter Search API, but in this case it is only possible to gather popular tweets published in the last seven days and there is a rate limit.

### 3.2.2 Aggregated analysis

5 An aggregated analysis of the sample of tweets was made through Gargantext, like for the article corpora. The term list (first row of key terms in Tab. S5.1, translated in Tab. S5.2) was established on the basis of the previous key term list and relevant new terms that are specific to Twitter jargon and also include English terms. The term resilience occurs fewer than five times also in this corpus. The occurrence of key terms was so limited that it was not possible to represent a significant graph, even after extracting terms referring to stakeholders and affected infrastructure. This is due to the limited number of characters that 10 are allowed for each tweet (up to 140 in 2016) and that make the information essential, unless the tweet includes a link to an external webpage (the content of which can't be automatically analysed through Gargantext).

Even if the thematic patterns of tweets could not be represented through a graph, we were able to push forward the aggregated analysis. We identified thematic groups of key terms and their frequency, as it was proposed by Vieweg et al. (2010).

### 3.2.3 Users' profile and behaviour

15 Besides this thematic analysis, the same sample of tweets can be used to investigate the behaviour of its users, their profiles and their interactions. We followed an approach which draws on the maps presented in the Climaps platform (Climaps, 2013). We identified the most liked, retweeted and active users in the sample. Indeed, these data reveal who are the most influential Twitter users that have the capacity to shape the social perception of risks and urban resilience.

We considered as the 'most active users' those accounts that published more than 10 tweets in one month. We identified as 20 the 'most liked users' those accounts that received more than 50 likes per tweet in one month. We named those accounts that received more than 50 retweets per tweet in one month the 'most retweeted users'.

We then examined the area of activity of these users. This was established on the basis of the description included in each user's account.

We then gave a closer look at the first five most active users. We aimed to observe if frequent tweeting had an impact in 25 terms of popularity, i.e. in terms of likes and retweets.

## 4 Results

### 4.1 Aggregated analysis of the press articles

Due to a very high inflow of publications in few days, especially in the case of the Seine River flood, the information in Fig. 1 is presented in a semi-log plot to make the information clearer.

During Seine River flood the press coverage peak was reached on the 3rd of June 2016 (310 articles, corresponding to 29 437 terms) (Fig. 1a). In the case of the Alpes-Maritimes flood the press coverage peak (see Fig. 1d) is smaller (108 articles on the 4th of October, corresponding to 14 772 terms). As shown in Fig. 1d, after the maximum peak (on the 4th of October) the number of terms per day decreases much faster than in the Paris region case study (Fig.1a). A peculiarity in Fig. 1d is that there are two small peaks on the 12th of October and then on the 28th of October.

316 key terms referring to flood resilience solutions (see Tab. S2.1 and S2.2) were extracted from the two corpora of articles: the corpus of articles covering the Seine River flood and the corpus of articles covering the Alpes-Maritimes flood. The key term 'resilience' ('résilience' in French) was automatically extracted by Gargantext but its occurrence was below five in both corpora, hence it was excluded from the list. The portion of listed key terms varies between 0% and 10% in the first corpus (Fig. 1a). In the second corpus the subset varies from 0% to 9% (Fig. 1d), which is close to the percentages of the previous case study.

## 4.2 Graph representation based on the articles on the Seine River flood

### 4.2.1 Visual observation of the graph

The first graph (Fig. 2) includes 254 nodes, 445 edges and six clusters that can be associated with six macro-themes:

1. The cluster named 'Monitoring system' that brings together topics such as 'sensors', 'data', 'modelling', 'observation', 'estimates', 'possible underestimations', as well as actors that include national authorities (the French Ministry of the Environment, the Minister of the Environment, Vigicrues) and researchers (IRSTEA, Vazken Andreassian);

2. The cluster named 'Warning system' that includes key terms such as 'red warning', 'warning', 'information', 'municipal plan', 'shelter', 'traffic' and actors, including government representatives (the French Minister of the Interior, the Prime Minister), a law enforcement institution (Prefecture), a transport company (SNCF), the national meteorological service (Météo France), an electric utility company (ERDF), vulnerable population (elders);

3. The cluster named 'Impact on population and infrastructure' that gathers topics such as 'rescue', 'closure', 'evacuation' of the 'museum', 'station', 'hospitals', 'transports', 'boats', 'cars', 'electric network', 'zouave' and actors such as local authorities (the Mayor of Paris, Departmental Council), cultural institutions ('Louvre', 'Grand Palais', National French Library), rescue services ('police', 'volunteers', 'civil protection', 'René Rabiant'), affected population ('tourists', 'tenants');

4. The cluster named 'Short-term response and insurance system' that brings together 'state of natural emergency', 'fire brigade', 'rescue services', 'stored art works', 'François Hollande', 'the National Assembly', 'inhabitants';

5. The cluster named 'Economic impact' with topics like 'damages', 'repair', 'farms', 'agricultural holdings', 'companies', 'market gardening', 'insurance', 'diagnosis' and actors that include the victims ('farmers', 'producers', 'victims', 'ship owners'), organisations representing them ('farmers' union') and insurers ('Bernard Spitz');

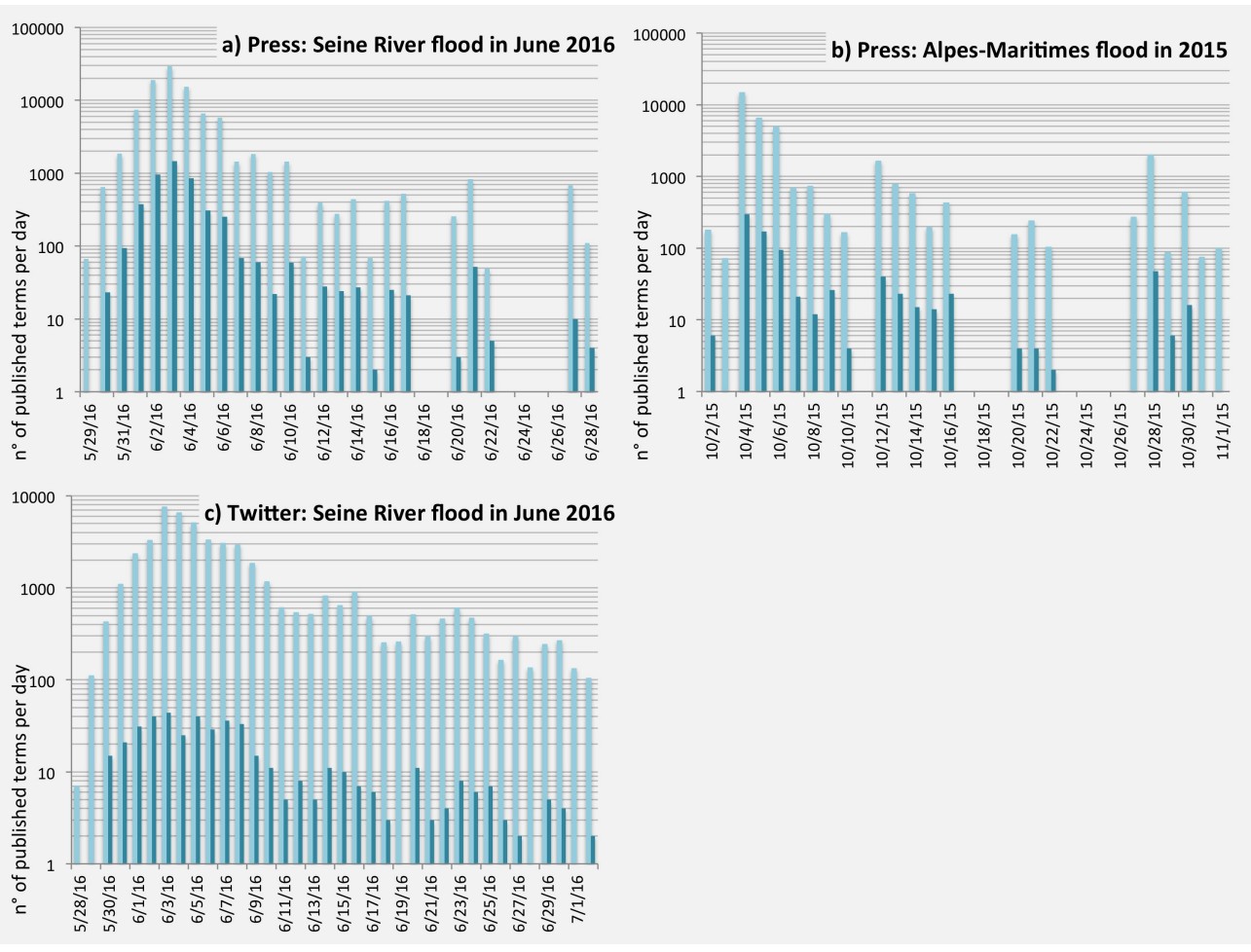

**Figure 1.** Comparison between the total number of terms per day (light blue) and the number of terms per day referring to flood resilience solutions (dark blue) in a semi-log plot (based on three different datasets).

6. The cluster named 'Long-term solutions' includes topics like 'awareness raising', 'prevention', 'soil sealing reduction', 'La Bassée pilot project', 'memory', 'preparation', 'first pilot area', 'new structure', 'public debate', 'retention basin', 'reinforce', 'simulation'; and actors such as local authorities ('Regional Department of Environment and Energy', 'Métropole du Grand Paris', 'associations', 'public territorial agencies', the mayors of Saint-Maur-des-Fossés and Rueil-Malmaison, the Hydrology Director at the Public Agency of the Seine Grands Lacs Basin).

### 4.2.2 Quantitative analysis of the nodes and the edges

As it is shown in Figure S3.1, the nodes with the highest degree (degree> 30) concern warning and emergency management, especially management of public infrastructure, indeed they are located in the two biggest clusters (the 'Warning system'

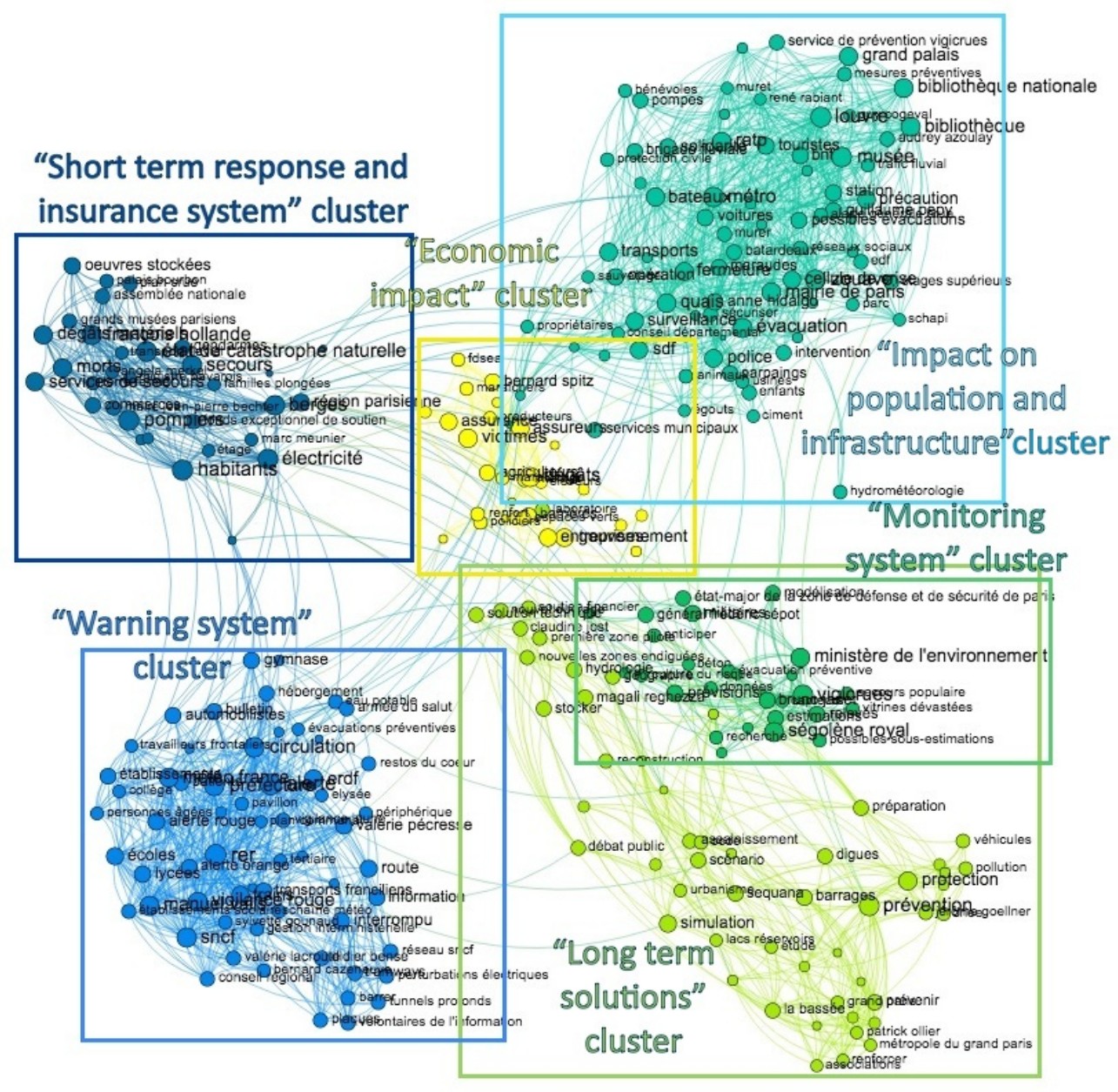

**Figure 2.** Graph representation computed with the press articles on the 2016 Seine River flood: co-occurrence graph computed on the basis of the measure of conditional proximity between pairs of key terms, in the corpus of articles covering the Seine River flood.

cluster and the 'Impact on population and infrastructure' cluster). For instance, the node with the highest degree is 'RER' (the

Paris region commuter rail service) with 55 edges. Concerning the actors, law enforcement and rescue services appear as the actors in the debate with the highest degree.

In Figure S3.2 the most probable co-occurrences (i.e. the highest edge weights) are identified with the label of the corresponding couple of nodes. The figure highlights which terms tend to be paired: terms that concern the same risk management field (e.g. rescue services); terms that concern the same area of affected activities (agriculture or transports); terms that concern infrastructure located in the same flood prone area; two terms that refer to an action and the object to which this action is directed (e.g. 'clubs' and 'closing') or a subject and an action performed by it (e.g. 'trains' and 'normally circulating').

## 4.3 Graph representation based on the articles on the Alpes-Maritimes flood

### 4.3.1 Visual observation of the graph

The second graph (Fig. 3) includes 104 nodes and 676 edges. As a consequence of the reduced key term occurrences, the number of nodes and edges is limited. Hence, not all the clusters are meaningful and can be identified with a macro-theme. However we identified the following four clusters with corresponding macro-themes:

1. A cluster of terms referring to the macro-theme 'Emergency management' that includes topics such as 'evacuation', 'rescue', 'hosted', 'hospitals', 'emergency procedures', and stakeholders such as 'prefecture', ' Alpes-Maritimes Prefecture', 'volunteers', the prime minister ('Manuel Valls'), the French president ('François Hollande'), the minister of the interior ('Bernard Cazeneuve'), 'the department council president';

2. A cluster of terms related to the macro-theme 'Monitoring system and prevention' with topics like 'orange warning' and stakeholders like the French national meteorological service ('Méteo France'), 'inhabitants';

3. A cluster of terms related to the macro-theme 'Reconstruction' that brings together 'insurance', 'insurers', 'compensation', 'accidents';

4. A cluster of terms related to the macro-theme 'Impact on population and infrastructure' that gathers terms such as 'traffic', 'shelter', 'deaths', 'missing (persons)', 'MIPCOM' (an international trade event that was disrupted).

Except for the macro-theme 'Impact on population and infrastructure', the other themes are not equivalent to those identified in the first graph.

### 4.3.2 Quantitative analysis of the nodes and the edges

The nodes with the highest degree (Fig. S4.1) correspond to the terms 'deaths', 'missing' and 'victims'. Furthermore, the following actors are among the high degree nodes: the government (including the Prime Minister and the French President), the inhabitants, the rescue services (the volunteers, the police, the fire brigade), the Cannes mayor, celebrities and insurers.

The values corresponding to the edge weights (Fig. S4.2) show that the most probable co-occurrences have some similarities with the trends described in the previous case study: some couple of terms concern the same area of flood resilience man-

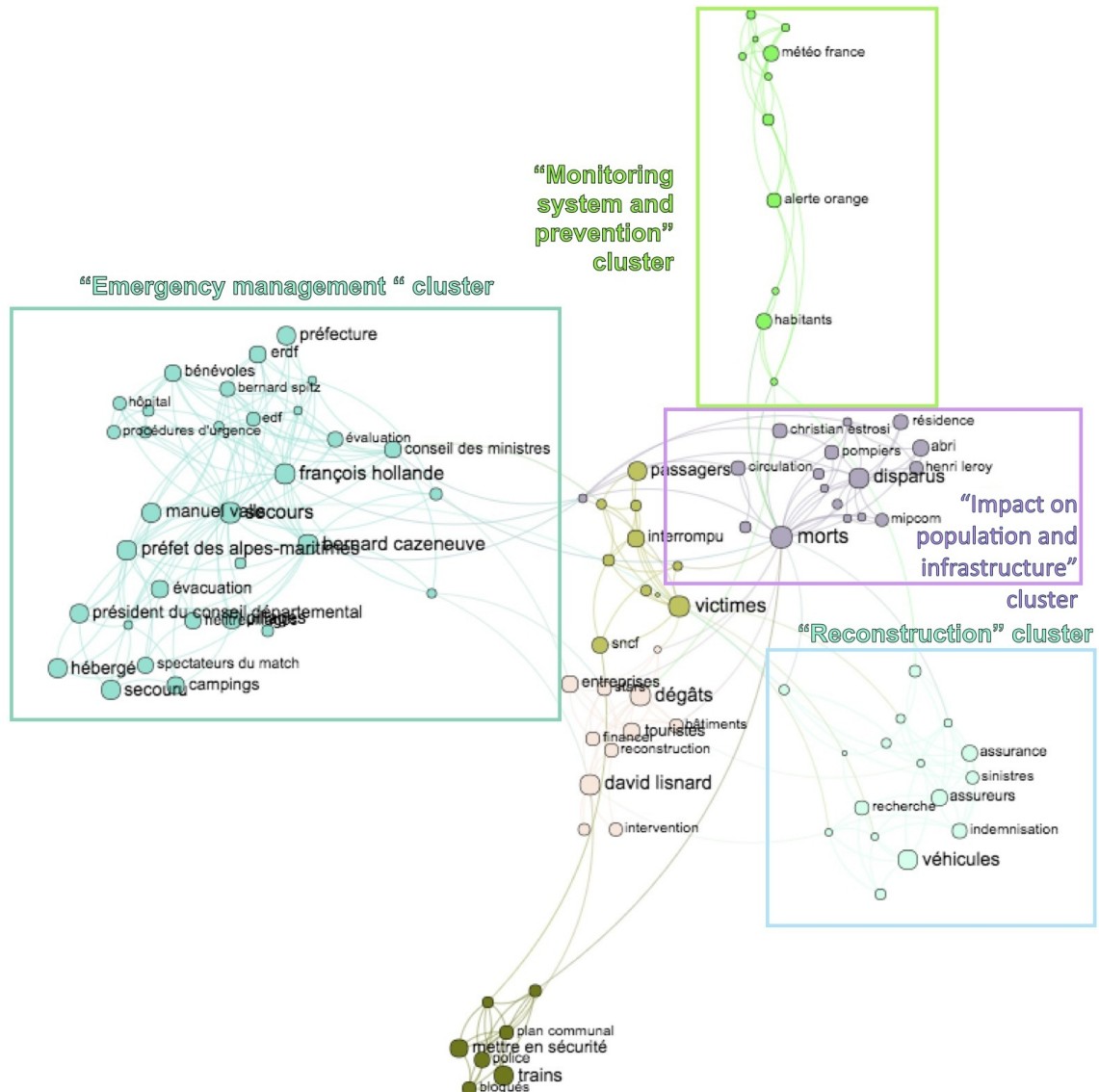

**Figure 3.** Graph representation computed with the press articles on the 2015 flood in the Alpes-Maritimes region: co-occurrence graph computed on the basis of the measure of conditional proximity between pairs of key terms in the corpus of articles.

agement (such as forecasts, impact report, awareness raising and prevention or compensations for the victims); other pairs of terms can be identified as an action and a related object (e.g. an event cancellation). Two actors that are frequently coupled are CNRS (the French National Centre for Scientific Research) and Météo France (the French national weather service). Two other frequent edges connect scientific organisations (IRSTEA and University of Avignon) with the topic 'awareness raising'.

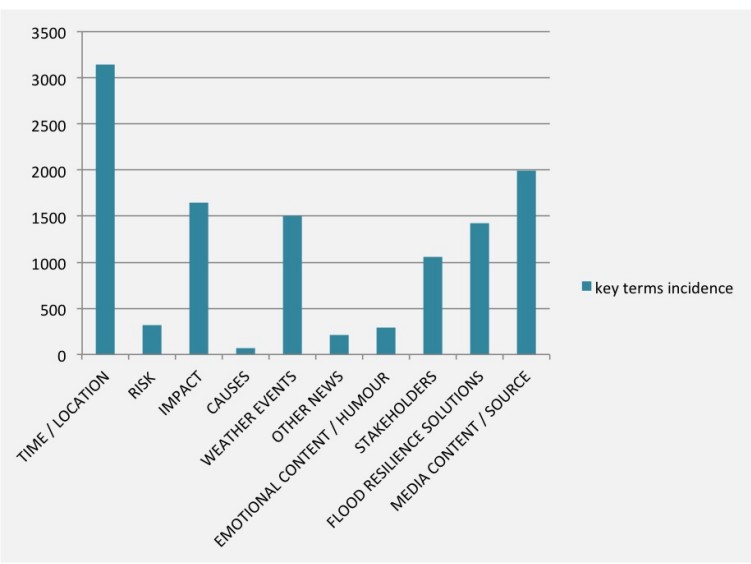

**Figure 4.** Twitter coverage of the 2016 Seine River flood: key terms incidence aggregated in ten thematic categories. The key terms were extracted from the corpus of tweets.

### 4.4 Aggregated analysis of the tweets on the Seine River flood

Figure 1c is based on a corpus of 7984 tweets and shows the number of terms published per day. The figure highlights that a term peak occurred on the 3rd of June 2016, like in the case of the press articles covering the same event. The dark blue columns in Fig. 1c represent the portion of terms referring to flood resilience solutions that varies between 0% and 3,5% of the
total number of terms published per day.

As shown in Fig. 4, a major portion of key terms (3143 occurrences) consists in purely factual information, with references to the time and location of the flood event. The category 'flood resilience solutions' (1420 occurrences) includes a relevant portion of terms, as well as the 'stakeholders' category (1060 occurrences). However these are less frequent than terms describing the weather event (1506 occurrences) and its impact (1644 occurrences). On the contrary, the debate on the causes (72 occurrences)
and threats (320 occurrences) is of little account.

### 4.5 Users' profile and behaviour

Figure 5 is based on the following sample: 59 'most active users' (who published more than 10 tweets in one month), 43 'most liked users' (who received more than 50 likes per tweet in one month) and 58 'most retweeted users' (who received more than 50 retweets per tweet in one month). In Figure 5a we compare individual accounts (i.e. accounts that belong to a person) with accounts that bear the name of an organisation in order to detect if followers react in a different way when they can
associate a tweet to a physical person. Among the most active users, 37% own an individual account. The most liked users are characterised by a majority of individual accounts (65%). The percentage is reduced in the case of the most retweeted users:

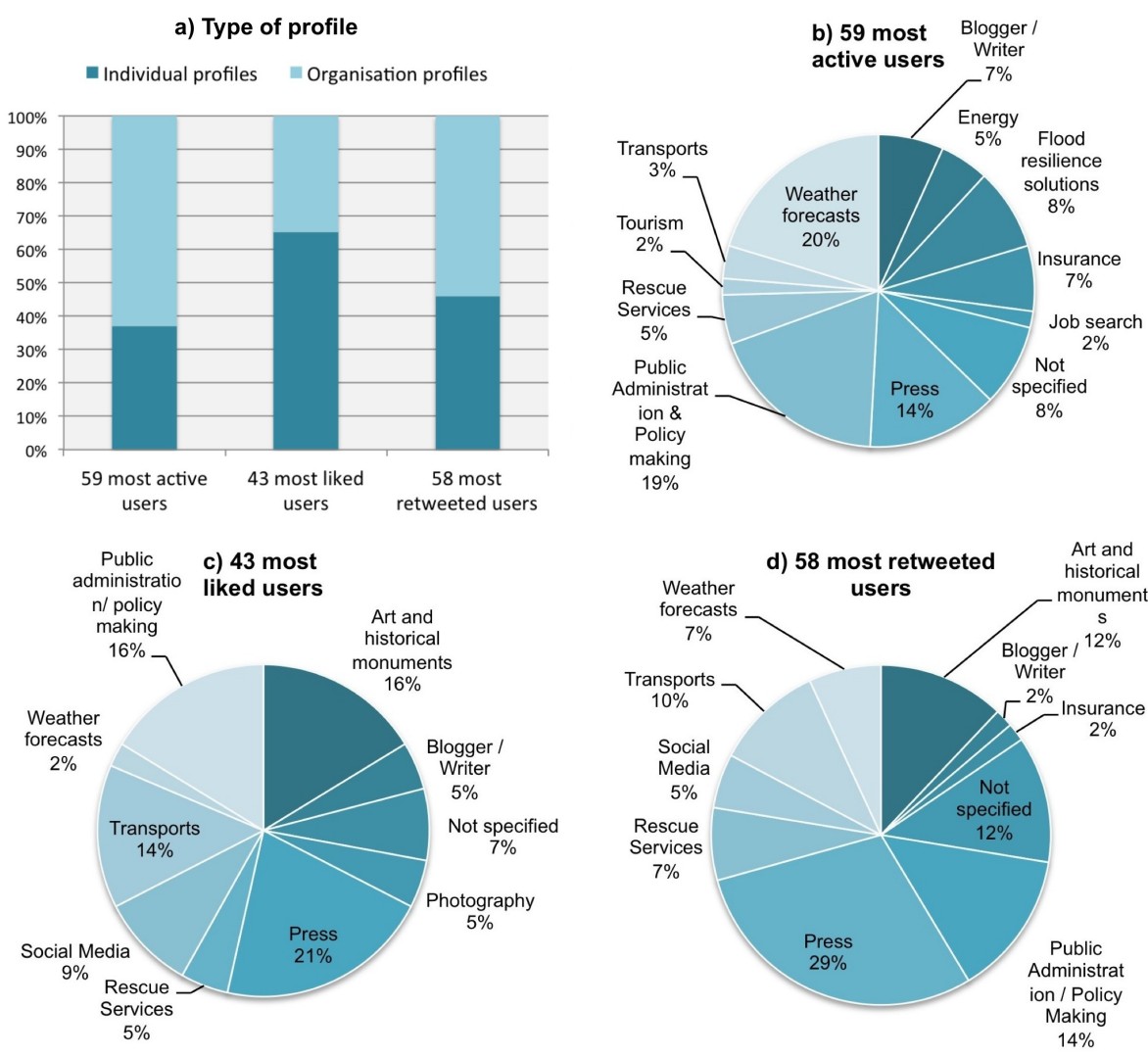

**Figure 5.** The users' behaviour (Twitter coverage of the 2016 Seine River flood): (a) percentage of individual profiles; (b) the area of activity of the most active users (59 users that published more than 10 tweets in one month); (c) the area of activity of the most liked users (43 users that received more than 50 likes per tweet in one month); (d) the area of activity of the most retweeted users (58 users that received more than 50 retweets per tweet in one month). These data were extracted from the tweet corpus and completed with information available on Twitter users' profile pages.

only 46% of them own an individual account. Hence, with a percentage difference of 19%, tweets from individuals generated less retweets than likes.

Figure 5b shows that the majority of users deal with weather forecasts (20%) or public administration/policy-making (19%), the next biggest area of activity gathers those users that are active in the field of journalism (14%). Information tweeted by public authorities and policy makers seems more frequent than information tweeted by rescue services (5%). As shown in Fig. 5 c, the most liked users are journalists (21%), followed by users that operate in the field of public administration/policy-making (16%), art/historical monuments (16%) and transports (14%). By looking at the areas of activity of the most retweeted accounts in Fig. 5d, we can notice similar trends to those presented in Fig. 5 c: the most relevant segment is the press (29%), followed by public authorities/policy-making (14%) and art/historical monuments (12%). The portion of tweets published by users that deal with transports is smaller (10%) than in Fig. 5 c with a percentage difference of 4%. Lastly, in Fig. S6, by putting in evidence the mean values and the upper bound[5] values, we can observe that the retweets and likes of the five most active users follow similar patterns.

## 5   Discussion

### 5.1   Comparison of the three histograms

A comparison of the three histograms in Fig. 1 reveals that, in the press as well as in social media, the peak of publications per day always occurs on the day of the highest river discharge. This clearly suggests a relation between a meteorological event and its social representation. A possible explanation could be that the press tends to rather focus on the immediate consequences of natural disasters (Houston et al., 2012). Twitter seems to follow the same trend as the press. Figure 1 also shows a noteworthy difference between the Alpes-Maritimes flood and the Seine River flood. The Seine River flood is marked by a slower decrease of press coverage, after the maximum coverage peak is reached. This kind of evolution could be a consequence of the very slow decrease of water levels of the Seine River, and it is probably reinforced by the high media visibility of Paris. Indeed, the events that occur in the French capital have a higher newsworthiness than those occurring in the rest of the country. Furthermore, the economic risks related to a flood event in Paris region are extremely high (OECD, 2014) since it is a densely populated area that represents a third of the national economy, and where companies' headquarters, national and international institutions are located; lastly, it is an important transportation node and one of the first tourist destinations in the world.

A peculiarity in Fig. 1d is that there are two small peaks: on the 12[th] of October this is caused by the press coverage of two rescued persons and two victims; on the 28[th] of October this is caused by the press coverage of the mayor of Cannes calling for help to the movie celebrities. It should also be noticed that in Fig. 1c the maximum peak is not as significant as in the histograms based on the press corpora. We suppose that Twitter can be influenced by the press agenda by it doesn't totally adhere to it.

The extraction of terms related to flood resilience solutions, through an iteration between manual analysis and automated text mining of the press corpora, revealed that the term 'resilience' occurs fewer than five times in each corpus: we suppose that in 2015 and 2016 the term resilience was not popular in the French media debate yet. Through an aggregated analysis of the list

---

[5]After finding the interquartile range (IQR) and the upper quartile (Q3), the upper bound is calculated with the following formula : Upper bound = Q3+1.5*IQR.

of key terms, we identified thematic subsets in each dataset and we observed the distribution of the number of published key terms per day (Fig. 1). The comparison of the three datasets calls attention to the minor discussion on flood resilience solutions on Twitter if we compare it to the debate in the press. Indeed Twitter is conducive to a fragmented communication and it is typically used as an early warning system: to disseminate factual information on the time and location of a flood.

## 5.2 Comparison of the graph representations

A comparison between the first and the second case study reveals some interesting similarities and differences between the two. The second corpus is characterised by a smaller occurrence of its key terms that generates a graph with a reduced number of edges and nodes. The macro-theme 'Impact on population and infrastructure' appears in both graphs and it is probably a recurring topic in the press coverage of natural disasters. The other macro-themes don't occur in both graphs. This is indicative of a certain variability between two cases of floods in terms of the resilience levers that are covered by the press.

In the first corpus of articles, the debate on various levers for flood resilience is well developed and detailed. Furthermore, specific stakeholders can be associated with specific resilience levers. The nodes with the highest degree in the first graph (for instance 'RER' with 55 edges) let us suppose that the revolved around immediate flood impacts the Seine River flood, i.e. how it affected a relevant portion of the population in their daily life. An unexpected result is that some stakeholders involved in flood risk management in Paris (e.g. public works companies, companies in charge of waste water collection) are not visible in the media debate in 2016. It is also surprising that the press doesn't refer to nature-based solutions as an alternative to traditional defence solutions to cope with flood risk.

The observation of the second graph let us suppose that the important number of victims drew the media attention to the tragic consequences of the flood event and to emergency management. Indeed, we observed that the stakeholders corresponding to high degree nodes are the affected population and those organisations that were involved in rescue activities, economic compensation and commemoration of the victims. Nevertheless, the most frequent edges reveal that part of the press debate revolved around the resilience solutions proposed by the scientists and the national weather service.

## 5.3 Analysis of the tweets

The aggregated analysis of thematic categories of Twitter terms highlighted that the most relevant portion of key terms consists in references to the time and location of the flood event. According to this result, Twitter might be primarily used as a means to disseminate warnings.

The analysis of Twitter profiles provided interesting insights on the most active users and the users that publish the most popular tweets. The percentages of individual profiles among the most popular users (Fig. 5 a) let us suppose that Twitter followers prefer supporting individual accounts by liking their tweets, while they tend to retweet less frequently. Twitter followers probably prefer to retweet from official sources rather than individuals for a reliability reason.

Another inference can be made on the basis of the profile of the users who published at least ten tweets in one month (Fig. 5b). A small percentage (5%) of profiles that belong to the rescue service are among the most active users. This could be explained by the fact that rescue services usually centralise information management.

By looking closer at the popularity of the five most active users (Fig. S8), we could observe that, in this small sample, there is no correlation between the number of tweets and the number of likes or retweets. However, if the number of likes is high, the number of retweets will probably be high as well.

By focusing on the tweets that obtained more than fifty likes or more than fifty retweets, it is possible to observe that the most popular tweets are published by the press and public authorities, i.e. those actors that are also visible in the press. Percentages presented in Fig. 5 c and Fig. 5d suggest that Twitter users operating in the media sector and in the public administration/policy-making sector are leading opinion makers in the debate on Paris flood risk. The press seems to raise broad interest: in the changing landscape of digital media, the press continues to be considered as a source of reliable information. Public authorities and policy makers are frequently in the social media spotlight, as their views can have direct consequences for the society. Twitter is a media open to any contributor. However, it seems that the most popular users are those actors that are also visible in the press. Fig. 5 c and 5d also call attention to a widespread interest among Twitter users in the flood impact on transports and cultural heritage: the inhabitants of Paris region, as well as tourists and people travelling across the region, felt strongly affected by the flood impacts on museums and transport infrastructure. This is probably specific to a population that lives or travels in a metropolis with a dense transport network and a very high concentration of historical monuments and museums.

Lastly, the portion of users dealing with 'transports' is smaller in the case of the 'most retweeted users' (Fig. 5d) than in the case of the 'most liked users' (Fig. 5 c). A percentage difference of 4% that could be explained by the fact that these tweets describe how transport workers cope with the flood, but they don't convey helpful information for the passengers.

## 6   Conclusions and perspectives

In this research we employed big data exploration techniques to investigate how urban resilience to extreme weather is perceived in the digital media debate. Through this study, we firstly intended to test how these techniques can be used to define indicators of social representation of urban resilience. In our view, these indicators can be integrated to a wider assessment of urban resilience to weather extremes. Secondly, we aimed at gaining an insight of the social perception of flood resilience in two French urban areas. This research is still in progress, however, through the experiments presented in this paper, we obtained quantitative data on:

– The relation existing between the intensity of the digital media debate (a social factor) and the level of the river discharge (an environmental factor);

– The evolution of the intensity of the debate over time, in two different locations (Seine River basin and Alpes-Maritimes Department) and in two different media contexts (French press and Twitter);

– The differences that exist in terms of quality of the content (i.e. reference to flood resilience solutions) between the press coverage of a flood event and the Twitter coverage of the same event;

– The topics and actors mentioned by the press that correspond to high degree nodes, and how these patterns change in two different urban areas;

- The most probable co-occurrences that exist among these topics and actors, and how these patterns change in two different urban areas;

- The most prominent topics in the Twitter debate;

- The profile of the most active and the most popular Twitter users.

The initial results are promising: these enable a complex understanding of the intensity and quality of the digital media debate. This research contributes to gain a better understanding of the public opinion, conveyed in the media, and the opinion makers which is beneficial for any urban resilience project in the Paris region. Indeed, this kind of analysis can contribute to creating a better connection with the urban community and optimise a project impact through dialogue and cooperation with the stakeholders.

In the future we intend to push forward our research by considering a longer time scale and by analysing the Twitter coverage of the 2015 Alpes-Maritimes flood. Furthermore, this methodology can be easily applied to other urban areas, affected by different climate related stresses and shocks. We also intend to study the correlations that might exist between the intensity and quality of the debate and other resilience variables, such as: the number of citizens affected by extreme weather, the surface of revegetated areas, the amount of insurance compensation for natural disasters, etc. Quantitative analysis of the graphs could be further developed: a detailed analysis of other measures that characterise graphs would be necessary to confirm our hypothesis based on the analysis of the node degrees and edges weight. Tweets analysis could be more fruitful if supported with automated exploration of Twitter accounts and graph representation of likes and retweets. It would be then possible to analyse larger samples of data and easily move from an aggregated level to a detailed level of analysis.

*Author contributions.* This research was carried out by RV and supervised by IT, BT and DS.

*Competing interests.* The authors declare that they have no conflict of interest.

*Acknowledgements.* The authors are thankful for the technical support provided by Institut des Systèmes Complexes Paris Île-de-France and Frédérique Bordignon from École des Ponts ParisTech during the implementation of Europresse, Gargantext and Gephi. The authors gratefully acknowledge the financial support of the chair 'Hydrology for resilient cities' endowed by Veolia.

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
