# Peer review of "Climate risks, digital media, and big data: following communication trails to investigate urban communities' resilience"

_Natural Hazards and Earth System Sciences, 2018_

## Referee Comment (RC1) · Anonymous Referee #1 · 4 Dec 2018

The work presents an interesting comparison of text analysis from news and Twitter, to identify urban resilience networks during flood events. The presented work and results are very interesting, but the paper needs to be organised differently and more technical details are necessary. Finally, a deeper analysis on why this work is useful needs to be presented. Comments: 1. Section 2.2 (Data) should, in my opinion, come before the Methods section (2.1), as the applied methods are specific to the collected data. 2. The Data section can be divided in 3 subsections to present the three datasets. 3. The Methods section should be considerably expanded. Most of the methodology is actually presented later on in the paper, and should be in this section instead. 4. It would be very helpful to explain how Gargantext algorithm works, what it is based on.

[Figure]

A lot is said about what Gargantext can do, but what did you do with it? Use active voices and present the logical order of passages. 5. Pg 3 Line 24: can you compare these statistics with the general population statistics? 6. How do you actually access the data from news and Twitter? Do you use an API? A scraper method? Which search criteria did you use? How many tweets did you download? You need all these details for reproducibility of results. A reader should be able to replicate all your steps. 7. Pg 4 Lines 28-30: these details should be in the data section. 8. Pg 4 Line 29: are these logical and/or? Is the and only between "inond*" and "Seine"? If so write is as an equation with correct parentesys. 9. Figure 1b: I would remove this panel. The case study is not presented in the analysis and generates confusion. 10. Pg 6 Lines 1-2: details about the zooming capabilities are not relevant. 11. Pg. 8 line 2: the colours are not relevant. Too much attention is given to the cluster colours, although this has been assigned without meaning. Please remove the sentence here and the colour references in the list below. It is also a limitation for colour-blinded readers. 12. Pg 8 Line 11: I would personally specify Social Impact. Similarly at line 18, I would call it Economic Impact. 13. No comment is done on the keyword "resilience", root concept in this paper. Is it find by the Gargantext networks? Is it common? 14. Figures S2.1 and S3.1: can you put all the keywords by the histogram? Just one out of two appears. 15. Pg 10 Lines 3-6: remove references to colours as they are not meaningful. 16. Figure 3: There is plenty of terms outside the defined clusters. Why the Impact Record cluster does not involve the keywords "passengers", "interrompu" and "victims" which seem relevant and close in the network? What about all the terms in the central/low part? 17. Pg 12 note 4: this should be included as a reference. 18. Pg 12 Lines 28-31: please explain why the "most liked users" and the "most retwitted users" are relevant in this analysis. What do they tell us about resilience? 19. Pg 13 Lines 4-5: Probably people prefer to retweet from official users/news rather than individuals for a reliability reason. You prefer to share info from an official source, rather than a person. 20. The Sections 3, 4, and 5 are already Results. I suggest you create a Section "Results" after "Methodology", with subsections for each of the case studies. Sections 3,4, and

5 also contain a lot of dscussions as well, which I would move to the "Results and Discussion" section, which should be renamed "Discussion" only. This would greatly improve the clarity of the manuscript. 21. Pg. 16 Line 27: the word "metric" would imply numerical values, but here you present mostly qualitative analysis. Do you have any plan to present additional quantitative analysis? 22. A big question is not answered: what is this study useful for? What can we learn from all this analysis? Why is it helpful? Is there anything that we can do differently in the future because of what we have learned?

---

## Referee Comment (RC2) · Anonymous Referee #2 · 11 Dec 2018

This paper describes a study on the content analysis of the press and tweets about floods in France. The topic is relevant to the journal and the study is of interests to international readers. I support its publication subject to the following improvements (mainly clarifications):

1) The study has used Gargantext and Gephi systems. Please explain why they were chosen? Are there any alternative systems that could also be used? 2) In "when few documents are deleted from the corpus or few nodes are removed from the network." Do you mean "a few" instead of "few"? 3) "excluding BTS"? what is "BTS"? 4) In "A corpus of 761 articles was first selected through Europresse archives", why did you

choose Europresse? Any other alternative sources? 5) "the following criteria: French press articles published from 15/05/2016 to 15/10/2016," why did you choose the 5 month duration? 6) In Figure 1, any explanations on the no values? 7) In "Table S1: Keywords related to flood resilient solutions", please explain the keywords in English and elaborate how they were selected (most readers cannot understand France, so it is useful to show the flood resilient solutions in English) 8) In Figure S2.1, please show both the French and English terms. 9) In "The corpus of tweets covering the Seine River flood of June 2016 was extracted through "Twitter Advanced Search" (twitter.com/searchadvanced)." Any alternative sites? Why this? 10) "The selection criteria were a time span (from 28/05/2016 to 2/7/2016)" why is such a duration chosen? 11) There are many grammatical errors. Please check through the whole manuscript to remove them (e.g., "it is possible to quickly navigating through"); "This is probably due to the higher newsworthiness that events in the French capital have in comparison to those occurring in the rest of the country." etc. . .).

---

## Author Comment (AC1) · 22 Jan 2019

The authors would like to thank the Reviewer for reviewing our manuscript and for providing the authors with their valuable remarks and recommendations. Furthermore, we are glad to see their recommendation for the publication of our manuscript. We have addressed all issues raised in their critique and we believe that our manuscript is now much stronger after addressing these comments. We hope that the proposed changes will satisfy his requirements. Here is a list of our preliminary responses to their comments:

Reviewer 2: This paper describes a study on the content analysis of the press and

tweets about floods in France. The topic is relevant to the journal and the study is of interests to international readers. I support its publication subject to the following improvements (mainly clarifications):

1)

R2: The study has used Gargantext and Gephi systems. Please explain why they were chosen? Are there any alternative systems that could also be used?

A: We agree with the Reviewer that this choice should be explained in the manuscript. We chose these two open-source software to facilitate the replicability of the study and because they both present some advantages. Gargantext is a network representation tool that is unique in terms of ergonomics. Indeed, it allows the analysis of text corpora at different levels: a micro level (the selection of key terms in a single document), a meso level (the selection of key terms and creation groups of terms in the table of terms extracted by Gargantext), a macro level (the selection of key terms in the network representation). Furthermore, thanks to the collaboration between the HM&Co laboratory and ISC-PIF we benefited of support from the developers of Gargantext. Concerning Gephi, it is a software that opens GEXF files, the format of the graph files produced by Gargantext. Furthermore, with Gephi it is possible to easily convert GEXF file in two Excel tables with the node degrees and the edge weights.

2)

R2: In "when few documents are deleted from the corpus or few nodes are removed from the network." Do you mean "a few" instead of "few"?

A: Thank you for pointing at this grammar mistake, we will correct it.

3)

R2: "excluding BTS"? what is "BTS"?

A: We agree with the Reviewer that this term should be explained. BTS (Brevet de

Technicien Supérieur) is a French diploma of higher education that is obtained after two or three years of studies in a highly specialised field.

4)

R2: In "A corpus of 761 articles was first selected through Europresse archives", why did you choose Europresse? Any other alternative sources?

A: We agree with the Reviewer that this choice should be explained in the manuscript. Europresse allows the export of press articles in a format that is compatible with Gargantext. Furthermore, Europresse gives access to press sources in different languages, unlike Argus de la Presse (another important press online archive). This characteristic opens the path to future research applied to other countries than French-speaking countries.

5)

R2: "the following criteria: French press articles published from 15/05/2016 to 15/10/2016," why did you choose the 5 month duration?

A: We carried out the analysis of the press articles on the Seine river flood five months after the flood. Following the Reviewer's comment, we will add this information in the manuscript.

6)

R2: In Figure 1, any explanations on the no values?

A: We agree with the Reviewer that adding the label "n° of published terms per day" beside the y-axis will facilitate the comprehension of the figure.

7)

R2: In "Table S1: Keywords related to flood resilient solutions", please explain the keywords in English and elaborate how they were selected (most readers cannot understand France, so it is useful to show the flood resilient solutions in English).

A: Following the Reviewer's comment, we will include an English translation of the selected terms and we will include the following comment: given that the objective of the analysis was to comprehend how Paris resilience is represented by the media, we selected all terms referring to solutions to cope with flood risk. For a better comprehension of the context where these solutions were implemented, we included in the selection the terms referring to Paris infrastructure. Lastly, since another objective of this analysis was to identify opinion makers, we also selected terms referring to stakeholders.

8)

R2: In Figure S2.1, please show both the French and English terms.

A: Following the Reviewer's comment, we will include an English translation of all the terms in the Figure.

9)

R2: In "The corpus of tweets covering the Seine River flood of June 2016 was extracted through "Twitter Advanced Search" (twitter.com/searchadvanced)." Any alternative sites? Why this?

A: We agree with the Reviewer that this choice should be explained. We will include the following comment: Twitter Advanced Search is a free and ergonomic service provided by Twitter. This online search tool, associated with Data Miner plug-in, allows the extraction of tweets even after their publication and without any rate limit. An example of a different method to extract tweets is Twitter Search API, but in this case it is only possible to gather popular tweets published in the last seven days and there is a rate limit.

10)

R2: "The selection criteria were a time span (from 28/05/2016 to 2/7/2016)" why is such a duration chosen?

A: We decided to consider the same duration in the analysis of the first and the second press article corpora in order to facilitate a cross comparison between the two case studies. Following the Reviewer's comment, we will add this information in the manuscript.

11)

R2: There are many grammatical errors. Please check through the whole manuscript to remove them (e.g., "it is possible to quickly navigating through"); "This is probably due to the higher newsworthiness that events in the French capital have in comparison to those occurring in the rest of the country." etc. . .).

A: Thank you for pointing at these mistakes, we will check the whole manuscript.

---

## Author Comment (AC2) · 22 Jan 2019

We would like to thank Reviewer 1 for his time to review our manuscript and to provide constructive comments and suggestions. We will address all issues raised in the critique and we believe that our manuscript will be much stronger after addressing these comments. Hopefully the implemented changes will satisfy his requirements. Here we would like to list our preliminary responses to the items raised by the Reviewer:

Reviewer 1: The work presents an interesting comparison of text analysis from news and Twitter, to identify urban resilience networks during flood events. The presented work and results are very interesting, but the paper needs to be organised differently

and more technical details are necessary. Finally, a deeper analysis on why this work is useful needs to be presented.

Authors: We appreciate that the Reviewer expressed his interest for the research presented in this paper. We agree with him that the manuscript could benefit of a better organisation of its contents, additions with technical and explanations on why this research is useful.

1.

R1: Section 2.2 (Data) should, in my opinion, come before the Methods section (2.1), as the applied methods are specific to the collected data.

A: Following the Reviewer's comment, we will invert the order of the two sections.

2.

R1: The Data section can be divided in 3 subsections to present the three datasets.

A: Following the Reviewer's comment, we will divide Data section in three subsections

3.

R1: The Methods section should be considerably expanded. Most of the methodology is actually presented later on in the paper, and should be in this section instead.

A: We thank the Reviewer for this suggestion, we will enrich the Methods section with information from the following sections and the details requested in comment #4.

4.

R1: It would be very helpful to explain how Gargantext algorithm works, what it is based on. A lot is said about what Gargantext can do, but what did you do with it? Use active voices and present the logical order of passages.

A: We agree with the Reviewer that further details on Gargantext algorithms would be useful. The first list of terms is automatically extracted by Gargantext algorithms

on the basis of their occurrence, compared to other occurrences that characterise all Gargantext database corpora, as well as on the basis of the co-occurrencies that characterise the specific corpus. Then, the network representations are computed on the basis of the Louvain modularity. The Louvain method for community detection is used to maximise the network modularity. A network with high modularity has dense edges between the nodes within modules and sparse edges between nodes belonging to different modules. The modularity maximisation involves two stages: first the small clusters are detected, then the nodes that belong to the same cluster are aggregated and a new network is produced whose nodes are the clusters. These operations are repeated until a maximum of modularity is reached and a clusters hierarchy is built. All other publicly available information on how Gargantext works is online in Gargantext documentation: https://iscpif.fr/gargantext/. Furthermore, we will specify in the paper that the tasks carried by the authors consisted of: 1) selecting and downloading the two corpora of press articles (news covering the 2016 Seine river flood and the news covering the 2015 Alpes-Maritimes flood) from Europresse in html format; 2) uploading the file on Gargantext; 3) selecting, among the terms automatically extracted by Gargantext from the two corpora, the most relevant terms and merge synonyms, declensions of terms and equivalent forms; 4) for each corpus analysis, keeping only those terms that occur at least 5 times; 5) verifying the daily distribution of terms referring to resilience solutions (through Gargantext Analytics view) and record the daily occurrence values in an Excel file to generate a histogram (fig. 1); 6) launching on Gargantext the network representation based on conditional distance among the selected list of terms; 7) launching, through the network visualisation engine, the algorithm that allows the strongly related nodes to be positioned close to each other; 8) selecting the option to visualise the terms corresponding to each node; 9) zooming in the network to observe all the nodes (even those with a small degree); 10) capturing a photo of all the networks; 11) extracting the network in .gexf format in order to analyse it with Gephi software; 12) importing the .gexf in Gephi and convert it in two Excel tables with the node degrees and the edge weight; 13) generating through Excel the figures presented

in the Supplement.

5.

R1: Pg 3 Line 24: can you compare these statistics with the general population statistics?

A: The authors thank the Reviewer for highlighting that this additional data are necessary. We will include these data in order to comprehend the differences between characteristics of the French population and of the Twitter users in France.

6.

R1: How do you actually access the data from news and Twitter? Do you use an API? A scraper method? Which search criteria did you use? How many tweets did you download? You need all these details for reproducibility of results. A reader should be able to replicate all your steps.

A: We agree with the Reviewer that further details are needed on the method that we employed to access the news and tweet data. Concerning the tweets, we have first selected the tweets through Twitter Advanced Searches on the basis of the following criteria: all tweets published between 28/05/2016 and 2/7/2016 and that contain at least one of the following hashtags: #crue, #crueparis, #crueseine, #inondation, #inondations, #pluies, #Seine. We then used a scraping tool, Dataminer (an open-source chrome extension software) that allows the conversion of HTML data that appear in the browser window into clean Excel table format. We first obtained 10073 tweets, this amount was reduced to 4497 after deleting the tweets referring to flood located in a different region than Île-de-France Region, the tweets that included the term #crue but referred to 'Motley Crue', 'uncooked food' or 'cruelty' in French.

Concerning the news, we accessed them through Europresse.com, a press online database that allows to select articles on the basis of keywords (in the title or in all the article), authors' name, language, type of media (frequency of distribution, geographical area of distribution, language, country), media name, publication dates. The selection criteria were: French press articles published from 15/05/2016 to 15/10/2016, with a title including the terms ("crue" or "inond*") and ("Seine" or "Ile-de- 30 France" or "Paris" or "Région Parisienne").

7.

R1: Pg 4 Lines 28-30: these details should be in the data section.

A: Following the Reviewer's comment, we will move these details to the data section.

8.

R1: Pg 4 Line 29: are these logical and/or? Is the and only between "inond*" and "Seine"? If so write is as an equation with correct parentheses.

A: We thank the Reviewer for suggesting the use of parentheses to facilitate the comprehension of the selection criteria. Parenthesis will be included as follows: ("crue" or "inond*") and ("Seine" or "Ile-de- 30 France" or "Paris" or "Région Parisienne")

9.

R1: Figure 1b: I would remove this panel. The case study is not presented in the analysis and generates confusion.

A: We agree with the Reviewer that it is better to remove Fig. 1b in order to improve the clarity of the paper.

10.

R1: Pg 6 Lines 1-2: details about the zooming capabilities are not relevant.

A: Following the Reviewer's comment, we will remove these details.

11.

R1: Pg. 8 line 2: the colours are not relevant. Too much attention is given to the

cluster colours, although this has been assigned without meaning. Please remove the sentence here and the colour references in the list below. It is also a limitation for colour-blinded readers.

A: We agree with the Reviewer that references to the colours in the text are not necessary. We will remove them.

12.

R1: Pg 8 Line 11: I would personally specify Social Impact. Similarly at line 18, I would call it Economic Impact.

A: We thank the Reviewer for suggesting these cluster names. We will use the title "Economic Impact" instead of "Affected market report", but we prefer to replace "Impact record" with "Impact on population and infrastructure" since some of the key terms included in this cluster refer to infrastructure.

13.

R1: No comment is done on the keyword "resilience", root concept in this paper. Is it find by the Gargantext networks? Is it common?

A: Following the Reviewer's remark, we will include the following comment: the key term "resilience" ("résilience" in French) was automatically extracted by Gargantext from the first corpus (news covering the Seine river flood) but its occurrence in was below 5. We suppose that in 2016 the term resilience was not popular in the media debate yet.

14.

R1: Figures S2.1 and S3.1: can you put all the keywords by the histogram? Just one out of two appears.

A: We thank the Reviewer for pointing at this inaccuracy. We will correct the two figures so that all the keywords are visible.

15.

R1: Pg 10 Lines 3-6: remove references to colours as they are not meaningful.

A: Following the Reviewer's comment, we will remove the references to colors.

16.

R1: Figure 3: There is plenty of terms outside the defined clusters. Why the Impact Record cluster does not involve the keywords "passengers", "interrompu" and "victims" which seem relevant and close in the network? What about all the terms in the central/low part?

A: Following the Reviewer's comment, we will specify that in the second network, as a consequence of the smaller term occurrences, the number of nodes and edges is limited. Hence, not all the clusters are meaningful and can be identified with a macrotheme. Concerning the three key terms mentioned by the Reviewer, they don't belong to the Impact Record cluster because they are not violet. Indeed Gargantext highlights with the same colour all the terms belonging to the same cluster.

17.

R1: Pg 12 note 4: this should be included as a reference.

A: Following the Reviewer's comment, we will include this reference.

18.

R1: Pg 12 Lines 28-31: please explain why the "most liked users" and the "most retwitted users" are relevant in this analysis. What do they tell us about resilience?

A: These data reveal which Twitter users are the most influential and have the capacity to shape the social perception of risks and of urban resilience. We thank the Reviewer for highlighting that this point was not clear, we will include this comment in the text.

19.

[Figure]

R1: Pg 13 Lines 4-5: Probably people prefer to retweet from official users/news rather than individuals for a reliability reason. You prefer to share info from an official source, rather than a person.

A: We thank the Reviewer for this pertinent remark that will be inserted in the manuscript.

20.

R1: The Sections 3, 4, and 5 are already Results. I suggest you create a Section "Results" after "Methodology", with subsections for each of the case studies. Sections 3,4, and 5 also contain a lot of dscussions as well, which I would move to the "Results and Discussion" section, which should be renamed "Discussion" only. This would greatly improve the clarity of the manuscript.

A: We agree with the Reviewer that these changes will improve the clarity of the paper. We will reorganise the information presented in Sect. 3, 4, 5 in a new "Results" section and in the "Discussion" section, as it is suggested by the Reviewer.

21.

R1: Pg. 16 Line 27: the word "metric" would imply numerical values, but here you present mostly qualitative analysis. Do you have any plan to present additional quantitative analysis?

A: We agree with the Reviewer that the term "metric" is not the most adequate, we will replace it with indicator. Additional quantitative analysis is planned as part of our future research.

22.

R1: A big question is not answered: what is this study useful for? What can we learn from all this analysis? Why is it helpful? Is there anything that we can do differently in the future because of what we have learned?

A: Following the Reviewer's remark, we will include a reflection based on the following answer: the results obtained through this research are relevant to gain a better understanding of the public opinion. These results will be beneficial for any urban resilience project in the Paris region. They will contribute to creating a better connection with the urban community and optimise the project impact trough dialogue and cooperation with the stakeholders. Furthermore, the methodology can be easily applied to other urban areas or different climate related stresses and shocks.

---

## Author Response (AR1)

First the authors would like to thank the Editor and the Reviewers for their time and comments that helped to improve the manuscript. We addressed all issues raised in the critique and we believe that our manuscript is now much stronger. Hopefully the changes implemented will meet their requirements! Here we list our answers to the items raised by Reviewer 1 and Reviewer 2.

**Answer to Reviewer 1**

*Reviewer 1: The work presents an interesting comparison of text analysis from news and Twitter, to identify urban resilience networks during flood events. The presented work and results are very interesting, but the paper needs to be organised differently and more technical details are necessary. Finally, a deeper analysis on why this work is useful needs to be presented.*

Authors: We appreciate that the Reviewer expressed his interest for the research presented in this paper. We agree with him that the manuscript needed:
—A better organisation of its contents. We address this issue in our answers to comments #1, #2, #3, #7, #20;
—Some additions with technical details. We address this issue in our answers to comments #4, #5, #6, #16;
—Explanations on why this research is useful. We address this issue in our answers to comments #18, #22.

*1.*

*R1: Section 2.2 (Data) should, in my opinion, come before the Methods section (2.1), as the applied methods are specific to the collected data.*

A: Following the Reviewer's comment, we inverted the order of the two sections.

*2.*

*R1: The Data section can be divided in 3 subsections to present the three datasets.*

A: Following the Reviewer's comment, we divided the Data section in the following three subsections:
>    *2.1 Press articles on the 2016 Seine River flood*
>    *2.2 Press articles on the 2015 flood in the Alpes-Maritimes Department*
>    *2.3 Tweets on the 2016 Seine River flood*

*3.*

*R1: The Methods section should be considerably expanded. Most of the methodology is actually presented later on in the paper, and should be in this section instead.*

A: We thank the Reviewer for this suggestion, we enriched the 'Methods "section with information from the sections 3, 4 and 5 of the first version of the manuscript. The new version of the 'Methods' section includes the following subsections:

*3.1 Method implemented to analyse the press articles*
*3.1.1 Dataset extraction*
*3.1.2 Aggregated analysis*
*3.1.3 Network representation*
*3.1.4 Visual observation of the network*
*3.1.5 Quantitative analysis of the nodes and the edges*
*3.2 Method implemented to analyse the tweets on the 2016 Seine River flood*
*3.2.1 Extraction of the dataset*
*3.2.2 Aggregated analysis*
*3.2.3 Users' profile and behaviour*

We also added further details on the methodology following comment #4 and #6. Please, see our answers to these comments.

*4.*

*R1: It would be very helpful to explain how Gargantext algorithm works, what it is based on. A lot is said about what Gargantext can do, but what did you do with it? Use active voices and present the logical order of passages.*

A: We agree with the Reviewer that the manuscript would benefit of further details on how Gargantext works and on what did the authors. We described these aspects in Supplement 1 'Details on how Gargantext works and step-by-step implementation'.

*5.*

*R1: Pg 3 Line 24: can you compare these statistics with the general population statistics?*

A: The authors thank the Reviewer for highlighting that this additional data are necessary in order to comprehend the differences between characteristics of the French population and of the Twitter users in France. We added these data in Footnote 2 (pag. 3).

*6.*

*R1: How do you actually access the data from news and Twitter? Do you use an API? A scraper method? Which search criteria did you use? How many tweets did you download? You need all these details for reproducibility of results. A reader should be able to replicate all your steps.*

A: We agree with the Reviewer that further details are needed on the method that we employed to access the press and tweet data.

Concerning the tweets, we included these details in Sect. 2.3: (from p. 4, line 20 to p. 5, line 2) and in 2.2.2.1 (p. 7, lines 21–24).
Concerning the press articles, we included more details on how Europresse works in Sect. 2.2.1.1 (p. 6, lines 6–8).

*7.*

*R1: Pg 4 Lines 28–30: these details should be in the data section.*

A: Following the Reviewer's comment, we moved all the details concerning the selection criteria of the press articles and tweets to the 'Data' section (see Sect. 2.1, 2.2, 2.3).

*8.*

*R1: Pg 4 Line 29: are these logical and/or? Is the and only between 'inond\*' and 'Seine'? If so write is as an equation with correct parentheses.*

A: We thank the Reviewer for suggesting the use of parentheses to facilitate the comprehension of the selection criteria. Parentheses were included as suggested at p. 3, lines 16–17 and at p. 4, lines 2–3.

*9.*

*R1: Figure 1b: I would remove this panel. The case study is not presented in the analysis and generates confusion.*

A: We agree with the Reviewer and we removed Fig. 1b in order to improve the clarity of the paper.

*10.*

*R1: Pg 6 Lines 1–2: details about the zooming capabilities are not relevant.*

A: Following the Reviewer's comment, we removed these details.

*11.*

*R1: Pg. 8 line 2: the colours are not relevant. Too much attention is given to the cluster colours, although this has been assigned without meaning. Please remove the sentence here and the colour references in the list below. It is also a limitation for colour-blinded readers.*

A: We agree with the Reviewer that references to the colours in the text are not necessary. We removed them.

*12.*

*R1: Pg 8 Line 11: I would personally specify Social Impact. Similarly at line 18, I would call it Economic Impact.*

A:  We thank the Reviewer for suggesting these cluster names. We replaced the title 'Affected market report' instead of 'Economic Impact', but we preferred to replace 'Impact record' with 'Impact on population and infrastructure' since some of the key terms included in this cluster refer to infrastructure. We made these changes in Fig. 2, Fig.3, p. 11 (lines 3 and 5), p. 13 (lines 9 and 10), p. 17 (line 4).

*13.*

*R1: No comment is done on the keyword 'resilience', root concept in this paper. Is it find by the Gargantext networks? Is it common?*

A: Following the Reviewer's remark, we included comments on this point in Sect. 4.1 (p. 8, lines 29–31), Sect. 5.1 (p. 16, lines 21–23).

*14.*

*R1: Figures S2.1 and S3.1: can you put all the keywords by the histogram? Just one out of two appears.*

A: We thank the Reviewer for pointing at this inaccuracy. We corrected the two figures  (S3.1 and S4.1 in the supplements to the new manuscript) so that all the keywords are visible.

*15.*

*R1: Pg 10 Lines 3–6: remove references to colours as they are not meaningful.*

A: Following the Reviewer's comment, we removed these references to colours.

*16.*

*R1: Figure 3: There is plenty of terms outside the defined clusters. Why the Impact Record cluster does not involve the keywords 'passengers', 'interrompu' and 'victims' which seem relevant and close in the network? What about all the terms in the central/low part?*

A: Following the Reviewer's comment, in Sect. 4.3.1 (p. 13, lines 3–4) we explain why some terms are outside the defined clusters.
Concerning the three key terms mentioned by the Reviewer, they don't belong to the 'Impact Record' cluster because they are not violet. In order to make sure that this point is clear, we specified in Sect. 3 (p. 6, lines 2–3) that 'Gargantext highlights with the same colour all the terms belonging to the same cluster'.

*17.*

A: Following the Reviewer's comment, we have included the note as the reference (Climaps, 2013) corresponding to 'Climaps: Reading the state of climate change from digital media, available at: http://disq.us/t/1gj2hci (last access: 18/03/2019), 2013.'

*18.*

*R1: Pg 12 Lines 28–31: please explain why the 'most liked users' and the 'most retwitted users' are relevant in this analysis. What do they tell us about resilience?*

A: We thank the Reviewer for highlighting that this point was not clear, we included the following explanation in Sect. 3.2.3 (p. 8, lines 10–11): 'Indeed, these data reveal which Twitter users are the most influential and have the capacity to shape the social perception of risks and of urban resilience'.

*19.*

*R1: Pg 13 Lines 4–5: Probably people prefer to retweet from official users/news rather than individuals for a reliability reason. You prefer to share info from an official source, rather than a person.*

A: We thank the Reviewer for this pertinent remark. We inserted it in Sect. 5.3 (p. 17, lines 25–26).

*20.*

*R1: The Sections 3, 4, and 5 are already Results. I suggest you create a Section 'Results' after 'Methodology', with subsections for each of the case studies. Sections 3,4, and 5 also contain a lot of dscussions as well, which I would move to the 'Results and Discussion' section, which should be renamed 'Discussion' only. This would greatly improve the clarity of the manuscript.*

A: We agree with the Reviewer that these changes will improve the clarity of the paper. As it is suggested by the Reviewer, we reorganised the information presented in Sect. 3, 4, 5 in a new 'Results' section and in the 'Discussion' section that include the following subsections:

*4 Results*

*4.1 Aggregated analysis of the press articles*

*4.2 Graph representation based on the articles on the Seine River flood*

*4.2.1 Visual observation of the graph*

*4.2.2 Quantitative analysis of the nodes and the edges*

*4.3 Graph representation based on the articles on the Alpes-Maritimes flood*

*21.*

*R1: Pg. 16 Line 27: the word 'metric' would imply numerical values, but here you present mostly qualitative analysis. Do you have any plan to present additional quantitative analysis?*

A: We agree with the Reviewer that the term 'metric' is not the most adequate, we replaced it with 'indicator' in the 'Conclusions and perspectives' section. We also specified that additional quantitative analysis is planned as part of our future research at p. 19, lines 12–13.

*22.*

*R1: A big question is not answered: what is this study useful for? What can we learn from all this analysis? Why is it helpful? Is there anything that we can do differently in the future because of what we have learned?*

A: Following the Reviewer's remark, we included the following reflections in the 'Conclusions and perspectives' section (p. 19, lines 4–9).

**Answers to Reviewer 2**

*Reviewer 2: This paper describes a study on the content analysis of the press and tweets about floods in France. The topic is relevant to the journal and the study is of interests to international readers. I support its publication subject to the following improvements (mainly clarifications):*

*23)*

*R2: The study has used Gargantext and Gephi systems. Please explain why they were chosen? Are there any alternative systems that could also be used?*

A: We agree with the Reviewer that this choice should be explained and we made related additions in Sect. 3 (p. 5, lines 18-21), Sect. 3.1.5 (p.7, lines 16-18) and in Supplement 1.

*24)*

*R2: In "when few documents are deleted from the corpus or few nodes are removed from the network." Do you mean "a few" instead of "few"?*

A: Thank you for pointing at this grammar mistake, we corrected it (in Supplement 1 on the new manuscript version).

*25)*

*R2: "excluding BTS"? what is "BTS"?*

A: We agree with the Reviewer that this term should be explained. We have included this information in Footnote 1 (p.3).

*26)*

*R2: In 'A corpus of 761 articles was first selected through Europresse archives', why did you choose Europresse? Any other alternative sources?*

A: We agree with the Reviewer that this choice should be explained in the manuscript. We have included this information in Sect. 3.1.1 (p. 6, lines 8–11).

*27)*

*R2: 'the following criteria: French press articles published from 15/05/2016 to 15/10/2016,' why did you choose the 5 month duration?*

A: Following the Reviewer's comment, we added the following information in Sect. 2.1. (p. 3, lines 17–19).

*28)*

*R2: In Figure 1, any explanations on the no values?*

A: We agree with the Reviewer: we added the label 'n° of published terms per day' beside the y-axis to facilitate the comprehension of the figure.

*29)*

*R2: In 'Table S1: Keywords related to flood resilient solutions', please explain the keywords in English and elaborate how they were selected (most readers cannot understand France, so it is useful to show the flood resilient solutions in English).*

A: Following the Reviewer's comment, we included in the Supplements an English translation of the selected terms (Tab. S3) and we explain why this selection was made in Sect. 3.1.2 (p. 6, lines 17–18) and Sect. 3.1.3 (from p. 6, line 31 to p. 7, line 3).

*30)*

*R2: In Figure S2.1, please show both the French and English terms.*

A: Following the Reviewer's comment, we have translated all the French terms in the Supplements.

*31)*

*R2: In 'The corpus of tweets covering the Seine River flood of June 2016 was extracted through "Twitter Advanced Search" (twitter.com/searchadvanced).' Any alternative sites? Why this?*

A: We agree with the Reviewer that this choice should be explained. We included this information in Sect. 3.2.1 (p.7, lines 21–26).

*32)*

*R2: 'The selection criteria were a time span (from 28/05/2016 to 2/7/2016)' why is such a duration chosen?*

A: We made an addition to explain this choice in Sect. 2.3 (p. 4, lines 21–23).

*33)*

*R2: There are many grammatical errors. Please check through the whole manuscript to remove them (e.g., 'it is possible to quickly navigating through'); '
[revised manuscript text omitted]

rosa 18/3/19 19:21

rosa 19/3/19 16:06

rosa 11/3/19 16:54
Moved up [2]: The peak of publis… [27]

rosa 11/3/19 20:13

rosa 11/3/19 20:17
Moved (insertion) [8]   [29]

rosa 18/3/19 19:26

rosa 18/3/19 17:00
Formatted   [31]

rosa 11/3/19 20:13

rosa 11/3/19 20:14
Moved down [7]: 1. A cluster of te… [33]

rosa 13/3/19 16:54

rosa 11/3/19 20:14
Moved (insertion) [7]   [35]

rosa 12/3/19 20:55

rosa 12/3/19 20:55

rosa 12/3/19 20:55

rosa 12/3/19 20:56

rosa 18/3/19 17:02
Formatted   [40]

rosa 13/3/19 15:59

rosa 18/3/19 19:21

rosa 18/3/19 17:02
Formatted   [42]

rosa 13/3/19 15:59

rosa 11/3/19 20:14

rosa 11/3/19 20:21
Moved down [9]: The nodes centr… [45]

rosa 13/3/19 17:26

rosa 11/3/19 20:21
Moved (insertion) [9]   [47]

rosa 15/3/19 13:05

rosa 18/3/19 17:03

[revised manuscript text omitted]

rosa 18/3/19 19:12

rosa 18/3/19 18:49

rosa 18/3/19 19:24

rosa 18/3/19 19:24

rosa 18/3/19 18:51

**Supplement S1: Details on how Gargantext works and step-by-step implementation.**

With Gargantext it is possible to extract, automatically as well as manually, a list of key terms from a corpus of digital texts. A first list of terms is automatically extracted by Gargantext algorithms on the basis of their occurrence, compared to other occurrences that characterise all Gargantext database corpora, as well as on the basis of the co-occurrences that characterise the specific corpus. This list can be manually modified: enriched with other terms extracted from the corpus, reduced, some terms can be merged in sub-lists. Gargantext is unique in terms of ergonomics. Indeed, it is possible to analyse of texts at different levels: a micro level (the manual selection of key terms in a single document), a meso level (the manual selection of key terms and creation of groups of terms in a table of terms extracted by Gargantext), a macro level (the manual selection of key terms in the graph representation).

The graph representations presented in this research are computed on the basis of the semantic proximity measure between pairs of key terms (from the list) that is called 'conditional distance'. Co-occurrence graphs based on conditional distance illustrate which terms co-occur in the same meaning unit (e.g. a press article) and with what probability. As it is specified in Gargantext documentation for advanced users (iscpif.fr/gargantext/mesures-utilisees-dans-gargantext/, last access: 18 March 2019), 'the conditional measure $P_c$ between term $i$ and term $j$ (...) is the maximum of the two conditional probabilities between $i$ and $j$. If $n_i$ ($n_j$ respectively) is the number of occurrences of $i$ ($j$ respectively) in the corpus of articles and $n_{ij}$ is the number of co-occurrences, we will have the following formula":

$$P_c = max\left(\frac{n_{ij}}{n_i}, \frac{n_{ij}}{n_j}\right)$$

Gargantext computes non-directed graph representations, i.e. these graph have no directed edges. The graph representations remain stable even when few documents are deleted from the corpus or few nodes are removed from the graph. Gargantext graphs are weighted a "weight" from zero to one is assigned to each edge of the graph and it indicates the probability that two terms co-occur. The degree of each node (i.e. the number of edges connected to the node) can be displayed through Gargantext graph visualisation engine.

The nodes are assembled in cohesive subsets through a clustering algorithm, more specifically through the Louvain modularity. The Louvain method for community detection is used to maximise the graph modularity. A graph with high modularity has dense edges between the nodes within modules and sparse edges between nodes belonging to different modules. The modularity maximisation involves two stages: first, the small clusters are detected, then the nodes that belong to the same cluster are aggregated and a new graph is produced whose nodes are the clusters. These operations are repeated until a maximum of modularity is reached and a clusters hierarchy is built.

All other publicly available information on how Gargantext works is presented in Gargantext documentation (iscpif.fr/gargantext/, last access: 18 March 2019).

This research was supported by Gargantext, but it required the following manual interventions that were carried out by the authors:

1) We selected and downloaded the two corpora of press articles (articles covering the 2016 Seine river flood and the articles covering the 2015 Alpes-Maritimes flood) from Europresse in HTML format;
2) We uploaded the file on Gargantext;
3) We selected, among the terms automatically extracted by Gargantext from the two corpora, the most relevant terms and we merged synonyms, declensions of terms and equivalent forms;
4) We kept only those terms that occur at least five times in one of the corpora;
5) We verified the daily distribution of terms referring to the resilience solutions (through Gargantext Analytics view) and we recorded the daily incidence values in an Excel file in order to generate a histogram (Fig. 1s);
6) We added to the key term list the terms referring to actors and affected infrastructure (with at least five occurrences) in order to represent in the graph the stakeholders involved in flood risk management and the context where the resilience solutions were implemented;
7) We launched on Gargantext the graph representation based on conditional distance between pairs of terms from the term list;

8) We launched, through the graph visualisation engine, the Louvain algorithm in order to position the strongly related nodes close to each other;

9) We selected the Gargantext option that displays the terms corresponding to each node and the node degrees;

10) We zoomed in the graph to observe all the nodes (even those with a small degree) and we captured a photo of the two graphs;

11) We extracted the graph in. gexf format in order to analyse it with Gephi software;

12) We imported the. gexf file in Gephi and we converted it in two Excel tables with the node degrees and the edge weights;

13) We generated through Excel the figures presented in the Supplements.
* * *
**Table S2.1: Key terms related to flood resilient solutions (in French). The terms were extracted from the articles on the 2016 Seine River flood and the 2015 Alpes-Maritimes flood. The list was automatically created by Gargantext algorithms, then it was manually refined on the basis of the relevance of the terms.**

[revised manuscript text omitted]

Adaptation, adapt, adapted, adaptations, adapted, alert, alerts, general flood alert, orange alert, orange alerts, orange vigilance, to alert, red alert, anticipate, insurance, insured, insurance, dams, dam, to bar, to block the access, gates, barriers, barrier, waterproof barriers, water barrier, water-barriers, water barrier, water barriers, stainless steel barriers, flood prevention barriers, storm water basin, storm water basins, locker, experimental, basins, retention reservoir, basin, retention reservoirs, pilot basin, cofferdams, cofferdam, bulletin, bulletins, sensors, measuring stations, hydroacoustics current meters, measuring equipment, hydro-acoustics current meters, hydrometric station, sensor, crisis cell, crisis centre, report, findings, cut, barred, risk culture, public debate, recovering trucks, recovering truck, increase permeable surfaces, revegetation, diagnoses, diagnosis, dikes, dyke, data, donations, lock, locks, closed school, closed schools, maintenance of streams, green spaces, green space, estimates, estimate, state of natural disaster, cat nat, aid plan, financial solidarity, support systems, natural disaster classification, natural disaster compensation fund, natural catastrophe guarantee, natural disaster procedure, solidarity system, solidarity fund, study, studies, evacuation, evacuations, preventive evacuation, assess, evaluation, assesses, evaluations, exceptional support fund, to train, management of aquatic environments and flood prevention, accommodation, shelter, emergency shelter, emergency shelters, shelter, relocated, to relocate, hosted, sheltering structures, hosts, accommodation, relocation, information, informed, to inform, prohibition, prohibited, prohibitions, forbidden, interrupted, total suspension, stopped, interrupted, ridge, interrupted, arrested, bassée, reservoir lakes, reservoir lakes, retention lakes, artificial lakes, four large lakes, lidar, memory, memories, safety measure, safety instructions, general safety rules, preventive measures, preventive measure, safety, to secure, secured, modelling, models, modelling, to wall up, to obstruct, low wall, low walls, new dyked areas, new construction, stored art pieces, concrete blocks, municipality plan, municipality plan of safeguard, municipality plans, municipality plans of safeguard, flood plan, prevention plans, risk prevention plans, prevention plan, preventive measures, natural risk prevention plans, flood risk prevention plans, prevention strategies, flood prevention action programs, flood risk prevention plans, protection plans, protection plan, plates, pumps, recovery pumps, to pump, pump, pumping, possible evacuation, precaution, precautions, first pilot area, preparation, prepare, prepared, prevent, prevention, preventive, preventives, precautionary, forecasts, forecast, predict, awareness, protection, protections, research, reconstruction, regulation, regulator, survey, automatic surveys, flow measurements, reinforce, reinforced, reinforcement, reinforcements, reparations, rebuild, rehabilitation, reparation, landmarks, landmark, restrictions, restriction, feedback, rescue, rescues, scenario, rescued, secure, awareness raising, awakened consciences, awareness of risk, Vigicrues prevention service, simulation, simulator, simulations, Vigiecrues site, solidarity, solidarities, technical solution, financial support, finance, financial aid, funds, repay, indemnified, compensate, emergency funds, grants, compensation payments, reimbursements, indemnifications, financial means, funds, financing, reimbursement, compensation, credit, financial assistance, grant, financing, finances, indemnified, storage, stocking, monitoring, surveillance, monitor, monitors, monitored, valve, valves, yellow vigilance, red vigilance red, steel shutters.

[Figure]

**Figure S3.1: The key terms corresponding to the nodes with the highest degree in the graph. The graph representation was computed on the basis of the measure of conditional proximity between pairs of key terms in the corpus of press articles on the Seine River flood.**

[Figure]

**Figure** S3.2: The most probable term co-occurrences corresponding to the edges with the highest weight in the graph. The graph is based on the corpus of press articles on the 2016 Seine River flood.

Unknown

[Figure]

**Fig. S4.1: The key terms corresponding to the nodes with the highest degrees in the graph. The graph representation is based on a corpus of press articles on the 2015 Alpes-Maritimes flood: it was computed on the basis of the measure of conditional proximity between pairs of key terms in the corpus of articles.**

[Figure]

**Fig.** S4.2: The most probable term key term co-occurrences corresponding with the highest weight in the graph. The graph is based on a corpus of press articles on the 2015 Alpes-Maritimes.

Unknown

**Table S5.1: Thematic groups of key terms based on the tweets on the 2016 Seine River flood. The key terms were extracted from the tweets corpus (in French).**

[revised manuscript text omitted]

| *Thematic group (or category)* | *Key terms extracted from the corpus of tweets* |
|---|---|
| FLOOD RESILIENCE SOLUTIONS | agents, operation, mobilised, exercises, we intervene, urgent, help, support, dam, naturaldisaster, crisis cells, insurance, vigilance, orange vigilance, Vigicrue, alert, red vigilance, firefighters, yellow vigilance, funds, police, prevention, security, red alert, rescue services, surveillance, call for donations, evacuation, are mobilised, orange alert, orange vigilance, PLOUF, urgent, simulation, CATNAT, precaution, volunteers, rescue, mobilisation, GEMAPI, reparations, concrete, PCSOrsay, recognition, works, mission, watergate, culture, aloa |
| MEDIA CONTENT / SOURCE | photos, periscope, information, photography, images, direct, videos, photographer, pic, view, news, artwork, streetphotography, maps, newspaper, pictures, photooftheday, nofilter, picoftheday, longexposure @ paris, photography https, latergram, image, infographic, news, press release, social media in emergency management, video, blog, dessin, qag, rediff, hashtag, www.youtube.com/watch, sky, Flickr, AFP, NouvelleRepublique, BFMTV, FRANCE2, SEPTAHUIT, Facebook |
| TIME/LOCATION | morning, this afternoon, last night, Monday, yesterday, tomorrow, Sunday, spring, departments, Nemours, municipalities, Essonne, city, Alma, Juvisy, Yvelines, Ile-de-France, Bercy, Europe, Montargis, Marne, Melun, city, Poissy, regions, Chatou, Austerlitz, Puteaux, ValdeMarne, Pontneuf, BoulogneBillancourt, Seine-et-Marne, Longjumeau, HautsdeSeine, Mirabeau bridge, Bièvre, Courbevoie, Alexandre III bridge, Issy, Cité, Jatte, Neuilly, Eiffel, Bastille, islands, Concorde, village, courthouse, sites, Paname, SeineRiverbnks, Yerres, Bougival, Saint Louis Island, ÎledelaCité, Grigny, countryside, Louvre, Almabridge, Suresnes, Georges Pompidou, Reuil, Mirabeau bridge, Alfortville, Asnieres, NotreDame, cathedral, Notre-Dame, Notre Dame, Paris Plage, Orsay, tunnel, castle, park, Lyon, statueofliberty, on the banks, arts |
| RISK | risk, flood-prone area, one-hundred-year flood, danger, historic flood, highly probable crisis, threat, be prudent, be careful |
| IMPACT | hit, disaster, cause, mud, stations, million euros, billion euros, flooded, charges, crisis, streets, ports, heritage, cars, boats, sculptures, difficulty, navigation, overflowed, overflows, evaluation of losses, dry, reopened, impacts, theSeineRiveroverflows, major flood, cut, blocked, bridge, disaster stricken population, traffic, museum, banks, damage, tourism, cultures, victims, RERC, closed, pollution, artwork, trains, house, death, agriculture, street, archive, cars, stations, transports, collection, traffic jam, expo, bill, cars, subway, accidents, station, cow, flooded, GDP, concert, waste water system, batobus, real estate, mosquito, restaurant |
| CAUSES | climatechange, climate, underestimated, urban development |
| WEATHER EVENTS | decrease of water levels, highest level, weather, rain, weather, record, measure, heights, unfetter, water levels, level, rise, seine decreases, meters, decreases, feet, seine level, weather, stabilisation, landmarks, comparison, meters, it goes up, hail, showers, deluge, normal levels, level, precipitations, Zouave, NASA, evolution, cumulation |
| OTHER NEWS | labourlaw, strike, SNCFstrike, notoDDL, strikes, terrorist attacks, Brexit, protests, migrants, CGT, nuitdebout |
| EMOTIONAL CONTENT/HUMOUR | parisIloveyou, thankful, love, braveness, impressive, fluctuatnecmergitur, hope, parisweloveyou, parismycity, thanks to, a thought for, boots, humour, swim, uber, unusual, pigeon, Venice, |
| STAKEHOLDERS | Hidalgo, individuals, inhabitants, council, Masson Delmotte, persons, City Hall, municipalities, populations, people, local authorities, affected municipalities, mayor, VLacroute, SleFoll, GrandParis, Seveso, Council of Ministers, Basin Comity, SNCF, Parisians, Hollande, Valls, children, RATP, companies, FRAECO, CCR, Courtier, clients, tourist, experts, TPE, AFA, IPRG, SMEs, farmers, Segolene |

[Figure]

[Figure]

**Figures S6: The users' behaviour: number of likes and number of retweets received by the five most active users for each of their tweets. These data were extracted from the sample of tweets on the 2016 Seine River flood.**

---

## Referee Report (RR1)

Title: Climate risks, digital media, and big data: following communication trails to investigate urban communities' resilience

Authors: Rosa Vicari, Ioulia Tchiguirinskaia, Bruno Tisserand, Daniel Schertzer

Summary and overview:

The aim of this study is to use web data produced during and after a flood to understand the interactions between the event and the social perception of the event using big data analysis techniques. The study looks at two different flood events in France (Alpes Maritimes 2015 and Paris 2016) and analyses media coverage of the event in French news articles and on Twitter. It demonstrates useful methods for understanding the public perception of flood events that could be helpful in improving the management of such events. This work is novel, relevant and worthwhile. However, we have suggested minor corrections mainly around including a definition of urban resilience and a few clarifications.

General comments:

Paragraphs should not be single sentences. The first sentence of the paragraph should set the topic of the paragraph, and the remaining sentences should develop the topic.

It would be clearer if figures that are central to the discussion are included within the article itself rather than as a supplement, if possible.

Urban resilience should be clearly defined, since the study is framed in the context of urban resilience assessment (see for example "Defining urban resilience: A review", Sara Meerow et al.). Similarly, some background on what urban resilience is should be included in the introduction. This can be used to explain why certain topics are expected to appear in the press before/during/after a flood event. Currently topics relevant to flood resilience are assumed implicitly (e.g. p.17 "it is interesting to note that there is no discussion of nature-based solutions") but there is no explicit justification for this.

Specific comments:

p.3: "Animals being destroyed" is not the common phrasing

p.3: A sentence introducing the Seine event in similar terms to the Alpes-Maritimes event would be useful. What was the driver of flooding, and from which date to which date was the area in fact flooded?

p.3: A different number of days before the event is used in each case, why is this? How are start and end dates for the analysis selected?

p.4: Not clear what exactly is meant by "This bond has been progressively fading". Does it mean that the power of the media to shape public opinion is decreasing?

p.4: Why is Twitter not also used for the Alpes-Maritimes event?

p.5 check grammar: toone

p.6: Is Europresse free to use?

p.7 section 3.1.5: What kind of quantitative analysis was done once the data was exported to Excel, what was the purpose of exporting the data?

p.8: Clarify "the peak of published terms per day is reduced"

p.8: Clarify "less progressively"

p.11: Is "red vigilance" is not commonly used in English, is there a better way to translate this?

p.13 section 4.3.1: It would be good to give examples of words that are in each of the clusters in the bullet points (like in section 4.2.1) especially since the figures are in French and will not be understandable by all the readers.

p.14 figure 4: Not clear how the distinction is made between terms relating to risk (which is normally defined as hazard x impact) and those relating to the weather events and impact. Also not clear what is meant by 'media content'.

p.14: Could individual accounts not also be those of professionals/individuals in positions of responsibility? In that case, how do you distinguish between individual accounts with private interests and those who speak from a professional role?

p.15 figure 5a: If I understood correctly the intention of Figure 5a), it might be clearer to show it as a stacked bar chart that sums to 100 % overall but with two separate colours, one for individual and one for company accounts (with a legend specifying which colour corresponds to which type of account)

p.16: Grammar point; "…not AS significant as"

p.17: After "an unexpected result is that some stakeholders involved in flood risk management in Paris are not visible in the media debate in 2016" please provide some examples of which stakeholders are missing.

p.18: perhaps "relation" would be more accurate than "correlation" to explain the comparison that is made between river discharge and the number of publications on the topic.

p.18 line 24: It might be clearer to add in brackets the two locations in to clarify that two independent comparisons are made, one between the two locations, and another between two different types of media in one of the two locations.

p.18 quality of the content (not pluralised)

---

## Author Response (AR2)

Many thanks to the Editor for guiding the review and to Reviewer 3 for the time devoted to the review and his final helpful comments. We have addressed all the issues raised by the Reviewer and we believe that we have met his requirements.

ANSWER TO REVIEWER 3

Reviewer 3: The aim of this study is to use web data produced during and after a flood to understand the interactions between the event and the social perception of the event using big data analysis techniques. The study looks at two different flood events in France (Alpes Maritimes 2015 and Paris 2016) and analyses media coverage of the event in French news articles and on Twitter. It demonstrates useful methods for understanding the public perception of flood events that could be helpful in improving the management of such events. This work is novel, relevant and worthwhile. However, we have suggested minor corrections mainly around including a definition of urban resilience and a few clarifications.

Authors: The authors would like to thank the Reviewer for his feedback and we are glad to read that he considers the study as novel, relevant and worthwhile and that, according to him, the proposed method is helpful. We consider that that the suggested corrections are relevant and we hope that we have met his expectations with the new version of the manuscript.

1)

General comments:

R3: Paragraphs should not be single sentences. The first sentence of the paragraph should set the topic of the paragraph, and the remaining sentences should develop the topic.

A: The authors agree with the Reviewer that paragraph organisation would benefit of some improvements. In the new manuscript we have improved the following paragraphs according to the Reviewer suggestions: p. 6 l. 13-19, p. 16 l. 1-10, p. 16  l. 24-28, p. 16 l. 29-p. 17 l.4, p. 17 l. 27-30.

2)

R3: It would be clearer if figures that are central to the discussion are included within the article itself rather than as a supplement, if possible.

A: The authors understand the Referee's concern but consider that the figures in the Supplement are not central to the discussion.

3)

R3: Urban resilience should be clearly defined, since the study is framed in the context

of urban resilience assessment (see for example "Defining urban resilience: A review", Sara Meerow et al.). Similarly, some background on what urban resilience is should be included in the introduction. This can be used to explain why certain topics are expected to appear in the press before/during/after a flood event. Currently topics relevant to flood resilience are assumed implicitly (e.g. p. 17, "it is interesting to note that there is no discussion of nature-based solutions") but there is no explicit justification for this.

A: The authors agree with the Reviewer that this additional information would improve the quality of the manuscript. We have included background information on the concept of resilience at p. 2 l. 4-7. The authors also thank the Reviewer for suggesting such a relevant reference, nevertheless our study is part of a PhD research that includes a review on the concept of resilience and its operational implications. This review is the object of the following paper by Vicari et al., 2019: " Assessing the impact of outreach strategies in cities coping with climate risks". We refer to this paper in the new version of the manuscript (p. 2 l. 7).

4)

R3: Specific comments: p. 3: "Animals being destroyed" is not the common phrasing

A: Following the Reviewer's comment, we have replaced "Animals being destroyed" with "slaughtered animals" (footnote 3 at p. 3).

p. 3: A sentence introducing the Seine event in similar terms to the Alpes-Maritimes event would be useful. What was the driver of flooding, and from which date to which date was the area in fact flooded?

A: Following the Reviewer's comment we included this information at p. 3 l. 12.

5)

R3: p. 3: A different number of days before the event is used in each case, why is this? How are start and end dates for the analysis selected?

A: We agree with the Reviewer that this point should be clarified. For both case studies, "the selection starts from the day of publication of the very first press articles" on each flood event. The selection ends five months after the first day of publications. We have included this information at p. 3 l. 21-24 and at p. 4 l. 8-10.

6)

R3: p.4: Not clear what exactly is meant by "This bond has been progressively fading". Does it mean that the power of the media to shape public opinion is decreasing?

A: Following the Reviewer's comment, we replaced this sentence with "the role of editors and journalists as exclusive news mediators has been progressively fading." (p. 4 l. 14-15).

7)

R3: p.4: Why is Twitter not also used for the Alpes-Maritimes event?

A: We thank the Reviewer for this pertinent suggestion. The main focus of this study is on the 2016 Seine River flood because of its exceptional media visibility (as discussed at p. 3 l. 16-18). However, we agree with the Reviewer that an analysis of the Twitter coverage of the Alpes-Maritimes flood would be relevant. Hence we mentioned it as part of our research perspectives at p. 19 l. 13-14.

8)

R3: p.5 check grammar: toone

A: Following the Reviewer's comment, we corrected this typographical error (p. 5-6 l. 6).

9)

p. 6: Is Europresse free to use?

A: We thank the Reviewer for pointing at this missing information. Europresse is not free, indeed it is an archive that includes articles that are accessible only to the newspapers' subscribers. We have clarified this point at p. 6 l. 13-14.

10)

R3: p.7 section 3.1.5: What kind of quantitative analysis was done once the data was exported to Excel, what was the purpose of exporting the data?

A: We agree with the Reviewer that this point should be clarified. In the new version of the manuscript we specified that "the values corresponding to the node degrees and to the edge weights can be easily extracted through Gephi, a graph visualisation software (gephi.com, last access: 18 May 2018), and then compared**" (p. .7 l. 20-22) and that "with Excel it is then possible to order these values in descending order and generate the figures in Supplement S3.1, S3.2, S4.1, S4.2." (p. 7 l. 24-25).

11)

R3: p.8: Clarify "the peak of published terms per day is reduced"

A: Following the Reviewer's comment, we replaced the sentence with "the press coverage peak is smaller" (p. 9 l. 2).

12)

R3: p.8: Clarify "less progressively"

A: Following the Reviewer's comment, we replaced "less progressively" with "much faster" (p. 9  l. 4).

13)

R3: p.11: Is "red vigilance" is not commonly used in English, is there a better way to translate this?

A: Following the Reviewer's comment, we replaced "red vigilance" with "red warning" (p. 9 l. 18).

14)

R3: p.13 section 4.3.1: It would be good to give examples of words that are in each of the clusters in the bullet points (like in section 4.2.1) especially since the figures are in French and will not be understandable by all the readers.

A: We agree with the Reviewer that this addition is necessary. We have included some examples at p. 12 from l. 13 to l. 22.

15)

R3: p. 14 figure 4: Not clear how the distinction is made between terms relating to risk (which is normally defined as hazard x impact) and those relating to the weather events and impact. Also not clear what is meant by 'media content'.

A: We thank the Reviewer for highlighting that these terms are unclear. We replaced 'risk' with 'threats' (p. 14 l. 10, Supplement S 5.1, S 5.2) and 'media content' with 'media content (video, photo...)' (Supplement S 5.1, S 5.2).

16)

R3: p. 14: Could individual accounts not also be those of professionals/individuals in positions of responsibility? In that case, how do you distinguish between individual accounts with private interests and those who speak from a professional role?

A: The authors agree with the Reviewer that this point might be unclear for the reader. Rather than being interested in distinguishing private accounts from professional accounts, we are interested in distinguishing accounts that belong to a physical person from accounts that bear the name of an organisation. In the new version of the manuscript we specified that "In Figure 5a we compare individual accounts (i.e. accounts that belong to a person) with accounts that bear the name of an organisation in order to detect if followers react in a different way when they can associate a tweet to a physical person." (p. 14 l. 14-16).

17)

R3: p.15 figure 5a: If I understood correctly the intention of Figure 5a), it might be clearer to show it as a stacked bar chart that sums to 100 % overall but with two separate colours, one for individual and one for company accounts (with a legend specifying which colour corresponds to which type of account)

A: The authors thank the Reviewer for this suggestion that will improve the clarity of the figure. We have replaced fig. 5a with the following one ( at p. 15):

[Figure]

18)

R3: p. 16: Grammar point; "...not AS significant as"

A: Following the Reviewer's comment, we replaced "not significant as" with "not as significant as" (p. 16 l. 26).

19)

p.17: After "an unexpected result is that some stakeholders involved in flood risk management in Paris are not visible in the media debate in 2016" please provide some examples of which stakeholders are missing.

A: Following the Reviewer's comment, we included the following examples: public works companies, companies in charge of waste water collection (p. 17 l. 15).

20)

R3: p.18: perhaps "relation" would be more accurate than "correlation" to explain the comparison that is made between river discharge and the number of publications on the topic.

A: Following the Reviewer's comment, we replaced "correlation" with "relation" (p. 16 l. 14 and p. 18 l. 25).

21)

R3: p.18 line 24: It might be clearer to add in brackets the two locations in to clarify that two independent comparisons are made, one between the two locations, and another between two different types of media in one of the two locations.

A: Following the Reviewer's comment, we added: (Seine River basin and Alpes-Maritimes Department) (p.18 l. 27-28).

22)

R3: p.18 quality of the content (not pluralised)

A: Following the Reviewer's comment, we replaced "contents" with "content" (p. 6 l. 32 and p. 18 l. 29).

[revised manuscript text omitted]

**Supplement S1: Details on how Gargantext V2 works and step-by-step implementation.**

With Gargantext it is possible to extract, automatically as well as manually, a list of key terms from a corpus of digital texts. A first list of terms is automatically extracted by Gargantext algorithms on the basis of their occurrence, compared to other occurrences that characterise all Gargantext database corpora, as well as on the basis of the co-occurrences that characterise the specific corpus. This list can be manually modified: enriched with other terms extracted from the corpus, reduced, some terms can be merged in sub-lists. Gargantext is unique in terms of ergonomics. Indeed, it is possible to analyse of texts at different levels: a micro level (the manual selection of key terms in a single document), a meso level (the manual selection of key terms and creation of groups of terms in a table of terms extracted by Gargantext), a macro level (the manual selection of key terms in the graph representation).

The graph representations presented in this research are computed on the basis of the semantic proximity measure between pairs of key terms (from the list) that is called 'conditional distance'. Co-occurrence graphs based on conditional distance illustrate which terms co-occur in the same meaning unit (e.g. a press article) and with what probability. As it is specified in Gargantext documentation for advanced users (iscpif.fr/gargantext/mesures-utilisees-dans-gargantext/, last access: 18 March 2019), 'the conditional measure $P_c$ between term $i$ and term $j$ (...) is the maximum of the two conditional probabilities between $i$ and $j$. If $n_i$ ($n_j$ respectively) is the number of occurrences of $i$ ($j$ respectively) in the corpus of articles and $n_{ij}$ is the number of co-occurrences, we will have the following formula":

$$P_c = max\left(\frac{n_{ij}}{n_i}, \frac{n_{ij}}{n_j}\right)$$

Gargantext computes non-directed graph representations, i.e. these graph have no directed edges. The graph representations remain stable even when few documents are deleted from the corpus or few nodes are removed from the graph. Gargantext graphs are weighted a "weight" from zero to one is assigned to each edge of the graph and it indicates the probability that two terms co-occur. The degree of each node (i.e. the number of edges connected to the node) can be displayed through Gargantext graph visualisation engine.

The nodes are assembled in cohesive subsets through a clustering algorithm, more specifically through the Louvain modularity. The Louvain method for community detection is used to maximise the graph modularity. A graph with high modularity has dense edges between the nodes within modules and sparse edges between nodes belonging to different modules. The modularity maximisation involves two stages: first, the small clusters are detected, then the nodes that belong to the same cluster are aggregated and a new graph is produced whose nodes are the clusters. These operations are repeated until a maximum of modularity is reached and a clusters hierarchy is built.

All other publicly available information on how Gargantext works is presented in Gargantext documentation (iscpif.fr/gargantext/, last access: 18 March 2019).

This research was supported by Gargantext, but it required the following manual interventions that were carried out by the authors:

1) We selected and downloaded the two corpora of press articles (articles covering the 2016 Seine river flood and the articles covering the 2015 Alpes-Maritimes flood) from Europresse in HTML format;
2) We uploaded the file on Gargantext;
3) We selected, among the terms automatically extracted by Gargantext from the two corpora, the most relevant terms and we merged synonyms, declensions of terms and equivalent forms;
4) We kept only those terms that occur at least five times in one of the corpora;
5) We verified the daily distribution of terms referring to the resilience solutions (through Gargantext Analytics view) and we recorded the daily incidence values in an Excel file in order to generate a histogram (Fig. 1s);
6) We added to the key term list the terms referring to actors and affected infrastructure (with at least five occurrences) in order to represent in the graph the stakeholders involved in flood risk management and the context where the resilience solutions were implemented;
7) We launched on Gargantext the graph representation based on conditional distance between pairs of terms from the term list;

8) We launched, through the graph visualisation engine, the Louvain algorithm in order to position the strongly related nodes close to each other;
9) We selected the Gargantext option that displays the terms corresponding to each node and the node degrees;
10) We zoomed in the graph to observe all the nodes (even those with a small degree) and we captured a photo of the two graphs;
11) We extracted the graph in. gexf format in order to analyse it with Gephi software;
12) We imported the. gexf file in Gephi and we converted it in two Excel tables with the node degrees and the edge weights;
13) We generated through Excel the figures presented in the Supplements.

**Table S2.1: Key terms related to flood resilient solutions (in French). The terms were extracted from the articles on the 2016 Seine River flood and the 2015 Alpes-Maritimes flood. The list was automatically created by Gargantext algorithms, then it was manually refined on the basis of the relevance of the terms.**

[revised manuscript text omitted]

Adaptation, adapt, adapted, adaptations, adapted, alert, alerts, general flood alert, orange alert, orange alerts, orange vigilance, to alert, red alert, anticipate, insurance, insured, insurance, dams, dam, to bar, to block the access, gates, barriers, barrier, waterproof barriers, water barrier, water-barriers, water barrier, water barriers, stainless steel barriers, flood prevention barriers, storm water basin, storm water basins, locker, experimental, basins, retention reservoir, basin, retention reservoirs, pilot basin, cofferdams, cofferdam, bulletin, bulletins, sensors, measuring stations, hydroacoustics current meters, measuring equipment, hydro-acoustics current meters, hydrometric station, sensor, crisis cell, crisis centre, report, findings, cut, barred, risk culture, public debate, recovering trucks, recovering truck, increase permeable surfaces, revegetation, diagnoses, diagnosis, dikes, dyke, data, donations, lock, locks, closed school, closed schools, maintenance of streams, green spaces, green space, estimates, estimate, state of natural disaster, cat nat, aid plan, financial solidarity, support systems, natural disaster classification, natural disaster compensation fund, natural catastrophe guarantee, natural disaster procedure, solidarity system, solidarity fund, study, studies, evacuation, evacuations, preventive evacuation, assess, evaluation, assesses, evaluations, exceptional support fund, to train, management of aquatic environments and flood prevention, accommodation, shelter, emergency shelter, emergency shelters, shelter, relocated, to relocate, hosted, sheltering structures, hosts, accommodation, relocation, information, informed, to inform, prohibition, prohibited, prohibitions, forbidden, interrupted, total suspension, stopped, interrupted, ridge, interrupted, arrested, bassée, reservoir lakes, reservoir lakes, retention lakes, artificial lakes, four large lakes, lidar, memory, memories, safety measure, safety instructions, general safety rules, preventive measures, preventive measure, safety, to secure, secured, modelling, models, modelling, to wall up, to obstruct, low wall, low walls, new dyked areas, new construction, stored art pieces, concrete blocks, municipality plan, municipality plan of safeguard, municipality plans, municipality plans of safeguard, flood plan, prevention plans, risk prevention plans, prevention plan, preventive measures, natural risk prevention plans, flood risk prevention plans, prevention strategies, flood prevention action programs, flood risk prevention plans, protection plans, protection plan, plates, pumps, recovery pumps, to pump, pump, pumping, possible evacuation, precaution, precautions, first pilot area, preparation, prepare, prepared, prevent, prevention, preventive, preventives, precautionary, forecasts, forecast, predict, awareness, protection, protections, research, reconstruction, regulation, regulator, survey, automatic surveys, flow measurements, reinforce, reinforced, reinforcement, reinforcements, reparations, rebuild, rehabilitation, reparation, landmarks, landmark, restrictions, restriction, feedback, rescue, rescues, scenario, rescued, secure, awareness raising, awakened consciences, awareness of risk, Vigicrues prevention service, simulation, simulator, simulations, Vigiecrues site, solidarity, solidarities, technical solution, financial support, finance, financial aid, funds, repay, indemnified, compensate, emergency funds, grants, compensation payments, reimbursements, indemnifications, financial means, funds, financing, reimbursement, compensation, credit, financial assistance, grant, financing, finances, indemnified, storage, stocking, monitoring, surveillance, monitor, monitors, monitored, valve, valves, yellow vigilance, red vigilance red, steel shutters.

[Figure]

**Figure S3.1: The key terms corresponding to the nodes with the highest degree in the graph. The graph representation was computed on the basis of the measure of conditional proximity between pairs of key terms in the corpus of press articles on the Seine River flood.**

[Figure]

**Figure S3.2: The most probable term co-occurrences corresponding to the edges with the highest weight in the graph. The graph is based on the corpus of press articles on the 2016 Seine River flood.**

[Figure]

**Fig. S4.1: The key terms corresponding to the nodes with the highest degrees in the graph. The graph representation is based on a corpus of press articles on the 2015 Alpes-Maritimes flood: it was computed on the basis of the measure of conditional proximity between pairs of key terms in the corpus of articles.**

[Figure]

**Fig. S4.2: The most probable term key term co-occurrences corresponding with the highest weight in the graph. The graph is based on a corpus of press articles on the 2015 Alpes-Maritimes.**

**Table S5.1: Thematic groups of key terms based on the tweets on the 2016 Seine River flood. The key terms were extracted from the tweets corpus (in French).**

[revised manuscript text omitted]

rosa 10/6/19 13:57

**Table S5.2: Thematic groups of key terms based on the tweets on the 2016 Seine River flood. The key terms were extracted from the tweets corpus. (English translation)**

| Thematic group (or category) | Key terms extracted from the corpus of tweets |
|---|---|
| FLOOD RESILIENCE SOLUTIONS | agents, operation, mobilised, exercises, we intervene, urgent, help, support, dam, naturaldisaster, crisis cells, insurance, vigilance, orange vigilance, Vigicrue, alert, red vigilance, firefighters, yellow vigilance, funds, police, prevention, security, red alert, rescue services, surveillance, call for donations, evacuation, are mobilised, orange alert, orange vigilance, PLOUF, urgent, simulation, CATNAT, precaution, volunteers, rescue, mobilisation, GEMAPI, reparations, concrete, PCSOrsay, recognition, works, mission, watergate, culture, aloa |
| MEDIA CONTENT (PHOTO, VIDEO...) / SOURCE | photos, periscope, information, photography, images, direct, videos, photographer, pic, view, news, artwork, streetphotography, maps, newspaper, pictures, photooftheday, nofilter, picoftheday, longexposure @ paris, photography https, latergram, image, infographic, news, press release, social media in emergency management, video, blog, dessin, qag, rediff, hashtag, www.youtube.com/watch, sky, Flickr, AFP, NouvelleRepublique, BFMTV, FRANCE2, SEPTAHUIT, Facebook |
| TIME/LOCATION | morning, this afternoon, last night, Monday, yesterday, tomorrow, Sunday, spring, departments, Nemours, municipalities, Essonne, city, Alma, Juvisy, Yvelines, Ile-de-France, Bercy, Europe, Montargis, Marne, Melun, city, Poissy, regions, Chatou, Austerlitz, Puteaux, ValdeMarne, Pontneuf, BoulogneBillancourt, Seine-et-Marne, Longjumeau, HautsdeSeine, Mirabeau bridge, Bièvre, Courbevoie, Alexandre III bridge, Issy, Cité, Jatte, Neuilly, Eiffel, Bastille, islands, Concorde, village, courthouse, sites, Paname, SeineRiverbnks, Yerres, Bougival, Saint Louis Island, ÎledelaCité, Grigny, countryside, Louvre, Almabridge, Suresnes, Georges Pompidou, Reuil, Mirabeau bridge, Alfortville, Asnieres, NotreDame, cathedral, Notre-Dame, Notre Dame, Paris Plage, Orsay, tunnel, castle, park, Lyon, statueofliberty, on the banks, arts |
| THREAT | risk, flood-prone area, one-hundred-year flood, danger, historic flood, highly probable crisis, threat, be prudent, be careful |
| IMPACT | hit, disaster, cause, mud, stations, million euros, billion euros, flooded, charges, crisis, streets, ports, heritage, cars, boats, sculptures, difficulty, navigation, overflowed, overflows, evaluation of losses, dry, reopened, impacts, theSeineRiveroverflows, major flood, cut, blocked, bridge, disaster stricken population, traffic, museum, banks, damage, tourism, cultures, victims, RERC, closed, pollution, artwork, trains, house, death, agriculture, street, archive, cars, stations, transports, collection, traffic jam, expo, bill, cars, subway, accidents, station, cow, flooded, GDP, concert, waste water system, batobus, real estate, mosquito, restaurant |
| CAUSES | climatechange, climate, underestimated, urban development |
| WEATHER EVENTS | decrease of water levels, highest level, weather, rain, weather, record, measure, heights, unfetter, water levels, level, rise, seine decreases, meters, decreases, feet, seine level, weather, stabilisation, landmarks, comparison, meters, it goes up, hail, showers, deluge, normal levels, level, precipitations, Zouave, NASA, evolution, cumulation |
| OTHER NEWS | labourlaw, strike, SNCFstrike, notoDDL, strikes, terrorist attacks, Brexit, protests, migrants, CGT, nuitdebout |
| EMOTIONAL CONTENT/HUMOUR | parisIloveyou, thankful, love, braveness, impressive, fluctuatnecmergitur, hope, parisweloveyou, parismycity, thanks to, a thought for, boots, humour, swim, uber, unusual, pigeon, Venice, |
| STAKEHOLDERS | Hidalgo, individuals, inhabitants, council, Masson Delmotte, persons, City Hall, municipalities, populations, people, local authorities, affected municipalities, mayor, VLacroute, SleFoll, GrandParis, Seveso, Council of Ministers, Basin Comity, SNCF, Parisians, Hollande, Valls, children, RATP, companies, FRAECO, CCR, Courtier, clients, tourist, experts, TPE, AFA, IPRG, SMEs, farmers, Segolene |

rosa 10/6/19 13:57

[Figure]

[Figure]

**Figures S6: The users' behaviour: number of likes and number of retweets received by the five most active users for each of their tweets. These data were extracted from the sample of tweets on the 2016 Seine River flood.**